# A spatiotemporal reconstruction of the *C. elegans* pharyngeal cuticle reveals a structure rich in phase-separating proteins

Muntasir Kamal[1,2†], Levon Tokmakjian[2,3†], Jessica Knox[1,2†], Peter Mastrangelo[1,2], Jingxiu Ji[1,2], Hao Cai[4], Jakub W Wojciechowski[5], Michael P Hughes[6], Kristóf Takács[7], Xiaoquan Chu[8], Jianfeng Pei[9], Vince Grolmusz[7], Malgorzata Kotulska[5], Julie Deborah Forman-Kay[4,10], Peter J Roy[1,2,3*]

[1]Department of Molecular Genetics, University of Toronto, Toronto, Canada; [2]The Donnelly Centre for Cellular and Biomolecular Research, University of Toronto, Toronto, Canada; [3]Department of Pharmacology and Toxicology, University of Toronto, Toronto, Canada; [4]Molecular Medicine Program, The Hospital for Sick Children, Toronto, Canada; [5]Wroclaw University of Science and Technology, Faculty of Fundamental Problems of Technology, Department of Biomedical Engineering, Wroclaw, Poland; [6]Department of Cell and Molecular Biology, St. Jude Children's Research Hospital, Memphis, United States; [7]PIT Bioinformatics Group, Institute of Mathematics, Eötvös University, Budapest, Hungary; [8]Center for Quantitative Biology, Academy for Advanced Interdisciplinary Studies, Peking University, Beijing, China; [9]Department of Computer Science and Technology, Tsinghua University, Beijing, China; [10]Department of Biochemistry, University of Toronto, Toronto, Canada

*For correspondence: peter.roy@utoronto.ca

†These authors contributed equally to this work

**Abstract** How the cuticles of the roughly 4.5 million species of ecdysozoan animals are constructed is not well understood. Here, we systematically mine gene expression datasets to uncover the spatiotemporal blueprint for how the chitin-based pharyngeal cuticle of the nematode *Caenorhabditis elegans* is built. We demonstrate that the blueprint correctly predicts expression patterns and functional relevance to cuticle development. We find that as larvae prepare to molt, catabolic enzymes are upregulated and the genes that encode chitin synthase, chitin cross-linkers, and homologs of amyloid regulators subsequently peak in expression. Forty-eight percent of the gene products secreted during the molt are predicted to be intrinsically disordered proteins (IDPs), many of which belong to four distinct families whose transcripts are expressed in overlapping waves. These include the IDPAs, IDPBs, and IDPCs, which are introduced for the first time here. All four families have sequence properties that drive phase separation and we demonstrate phase separation for one exemplar in vitro. This systematic analysis represents the first blueprint for cuticle construction and highlights the massive contribution that phase-separating materials make to the structure.

## Editor's evaluation

Cuticles are specialized extracellular matrices that cover the bodies of ecdysozoans, which make up 85% of all animals, and how cuticles are formed is very poorly understood, in particular in light of the fact that cuticles are shed and regrown as animals grow. The authors present a comprehensively and carefully curated resource of the components of the pharyngeal cuticle of *C. elegans* and provide a

spatiotemporal framework to understand cuticle assembly. In doing so, the authors propose a function for a large class of intrinsically disordered proteins (IDPs). The significance of this work is high because our understanding of both cuticle formation and of IDPs is poor.

## Introduction

Over 85% of living animal species belong to the superphylum ecdysozoa. This group includes nematodes, arthropods, tardigrades, and five other phyla (*Telford et al., 2008*; *Aguinaldo et al., 1997*). They are defined by having a common ancestor and a specialized extracellular matrix that covers their body called the cuticle. The ecdysozoan cuticle is shed and regrown to accommodate juvenile growth in a process called ecdysis or molting.

Cuticle shape is patterned by the tissue beneath it, but also takes on additional diversity beyond the underlying tissue shape. One example of this structural diversity is the mouthparts of nematodes. Many carnivorous nematodes and nematode parasites of animals have cuticle-based teeth that bite into their prey or host (*Sieriebriennikov and Sommer, 2018*; *John and Petri, 2006*). Nematode parasites of plants have needle-like cuticle stylets that pierce plants and act as a syringe to deposit effectors and suck out vital nutrients (*Mejias et al., 2019*). Bacterivorous nematodes, like the model nematode *Caenorhabditis elegans*, have cuticle grinders that pulverize bacteria into digestible bits (*Sparacio et al., 2020*). These specialized mouthparts are variations of the cuticle that lines the anterior alimentary tract. Despite this diversity in form and the importance of the cuticle to most animals, a spatiotemporal blueprint for cuticle construction is lacking. Here, we provide such a blueprint by mining published datasets of *C. elegans* gene expression.

All epithelia in *C. elegans* that would otherwise be exposed to the environment, except the intestine, are protected by a cuticle. These include the body cuticle that protects the hypodermis (aka epidermis), the anterior alimentary cuticle that reinforces the lumen of the buccal cavity and pharynx, and other cuticles that protect the rectum, vulva, and excretory pore tissues (*Altun and Hall, 2020*). Here, we will refer to the anterior alimentary cuticle as the pharyngeal cuticle.

The non-chitinous body cuticle has multiple layers that include an outer carbohydrate-rich glycocalyx, a lipid-rich epicuticle, and multiple inner collagenous layers (*Altun and Hall, 2020*; *Page and Johnstone, 2007*; *Cox et al., 1981*). By contrast, the pharyngeal cuticle is not collagenous (*Altun and Hall, 2020*; *Cox et al., 1981*) and instead contains a chitin-chitosan matrix that likely helps maintain luminal integrity (*Zhang et al., 2005*; *Heustis et al., 2012*). The pharyngeal cuticle is layered (*Sparacio et al., 2020*; *Wright and Thomson, 1981*), but the molecular composition of the different layers is unknown. Like other ecdysozoans, *C. elegans* sheds its cuticles at the end of each larval stage. As the old cuticle is being shed, a new cuticle is built underneath, and the next developmental stage ensues (*Sparacio et al., 2020*; *Lazetic and Fay, 2017*). *C. elegans* adults do not molt.

In addition to chitin, the pharyngeal cuticle contains a group of largely disordered proteins called the APPGs (also known as the ABU/PQN Paralog Group) (*George-Raizen et al., 2014*). The APPGs are low complexity (i.e., they have a biased composition involving a limited set of amino acids) and have been described as prion-like (*Michelitsch and Weissman, 2000*) and potentially amyloidogenic (*George-Raizen et al., 2014*). An examination of the expression pattern of five APPGs showed that all five are expressed in cells that surround the pharyngeal cuticle and that APPG::GFP fusion proteins are incorporated into the pharyngeal cuticle (*George-Raizen et al., 2014*). The disruption of two of these genes exhibits feeding phenotypes consistent with disruption of this cuticle (*George-Raizen et al., 2014*). In this study, we find the APPGs to be one of several groups of proteins dominated by large intrinsically disordered regions (IDRs) with low-complexity sequences that are likely secreted into the developing pharyngeal cuticle.

IDRs are defined here as a 30 or more continuous residues whose primary sequence fails to form a stereotypical stable tertiary structure and instead rapidly interconverts between heterogenous conformations (*van der Lee et al., 2014*). Despite lacking ordered structure, IDRs can interact with other IDRs through local areas of hydrophobicity, complementary charge, hydrogen-bond formation, and pi-stacking interactions along the respective peptide chains (*Vernon and Forman-Kay, 2019*). IDRs often harbor repeating sequence features that can facilitate the formation of multivalent interaction networks with multiple binding partners (*Vernon and Forman-Kay, 2019*). Depending on the local environment, multivalent IDRs, and particularly low-complexity IDRs, can phase separate to

form liquid–liquid phase-separated droplets (LLPS) (i.e., liquid condensates) or gels, which can then transition to more solid structures, including fibers (*Mittag and Parker, 2018*; *Banani et al., 2017*). LLPS has been shown to be an important first step in the self-assembly of IDR-rich proteins into the extracellular matrices of insects, arachnids, and molluscs (reviewed in *Muiznieks et al., 2018*). For example, IDR-rich proteins that form liquid condensates fill a porous chitin-based matrix in a key step of squid beak development (*Tan et al., 2015*). Given that the affinity of any one interaction along an IDR is relatively weak, the ability of IDRs to form these phase-separated networks is easily modulated by a variety of factors, including pH, ions, temperature, protein concentration, and post-translational modifications (*Murray et al., 2017*).

Here, we describe the spatiotemporal logic of pharyngeal cuticle construction that we have uncovered by mining published mRNA expression datasets and canonical amyloid and chitin-binding dyes. We identify six families of low-complexity proteins that are likely secreted into the developing cuticle, including the IDPAs, IDPBs, and IDPCs, each of which are described for the first time here, and the APPGs, NSPBs, and the FIPRs. These six families peak in expression level in successive waves over the course of each larval stage. Computational analyses predict that the IDPA, IDPB, IDPC, and APPG families, and 12 other singletons are IDR-rich proteins capable of phase separation. We speculate that the malleable properties of the disordered phase-separating proteins are especially suited to a flexible cuticle that must be rapidly destroyed and reconstructed during molting.

## Results

### Validating fluorescent dyes as probes of pharyngeal cuticle structure

Earlier transmission electron microscopy of the *C. elegans* pharynx cuticle revealed it to be a complex structure that changes in character along its anterior–posterior axis (*Sparacio et al., 2020*; *Wright and Thomson, 1981*; *White et al., 1986*; *Figure 1*). To further characterize its structure, we first sought to validate dyes as probes of the cuticle. Congo Red (CR) fluoresces red and binds to amyloid oligomers, protofibrils, and fibrils (*Bennhold, 1922*; *Wu et al., 2012*) and has been previously shown to stain the cuticular grinder of the pharynx (*George-Raizen et al., 2014*). Thioflavin S (ThS) increases in blue fluorescence emission upon binding amyloid structures (*Vassar and Culling, 1959*). Calcofluor white (CFW) fluoresces deep blue and is used as a chitin probe in other systems (*Roncero et al., 1988*). Eosin Y (EY) is a yellow-red fluorescent dye that binds chitosan, which is the deacetylated form of chitin (*Baker et al., 2007*).

We confirmed that the four dyes specifically bind components within the pharyngeal cuticle in two ways. First, we performed pulse-chase experiments with the dyes to determine whether the dye's fluorescent signal would be lost as the larvae shed their old cuticle during their transition to the next developmental stage (see 'Materials and methods' for details). After the 18 hr chase, very few animals who were initially L3s had CFW, EY, CR, or ThS signal (*Figure 2*, *Figure 2—figure supplement 1*). By contrast, the dyes' signal persisted in animals that were initially young adults (*Figure 2*). The loss of the four dyes from the larvae but not adults in the pulse-chase experiments indicates that the dyes bind the pharyngeal cuticle.

Second, we tested whether the dyes bind the pharyngeal cuticle after the cuticle has separated from the animal, the attachment of which persists in *mlt-9(RNAi)* mutants (*Frand et al., 2005*). We found that all four dyes bind the exterior pharyngeal cuticle of *mlt-9(RNAi)* animals (*Figure 2S–X*). As a positive control, we find that GFP-tagged ABU-14 is retained in the shed pharyngeal cuticle (*Figure 2Y*). These data establish CR, ThS, CFW, and EY as specific probes of the pharyngeal cuticle.

### Cuticle dyes stain distinct structures within the pharyngeal cuticle

We examined the colocalization of the four dyes in wildtype animals and correlated the resulting patterns to the ultrastructural features observed in a series of unpublished TEM images by Kenneth A. Wright and Nicole Thomson (*Wright and Thomson, 1981*; *Figure 3*). These TEM images show that the cuticle of the buccal cavity and the channels is a mixture of electron-light and electron-dense (dark) material, with the dark material forming circumferential ribs (white arrows) and 'flaps' (yellow arrows).

Two features suggest that the chitin-binding dyes may bind components within the electron-light material. First, the expansive electron-light material at the anterior half of the buccal cuticle correlates

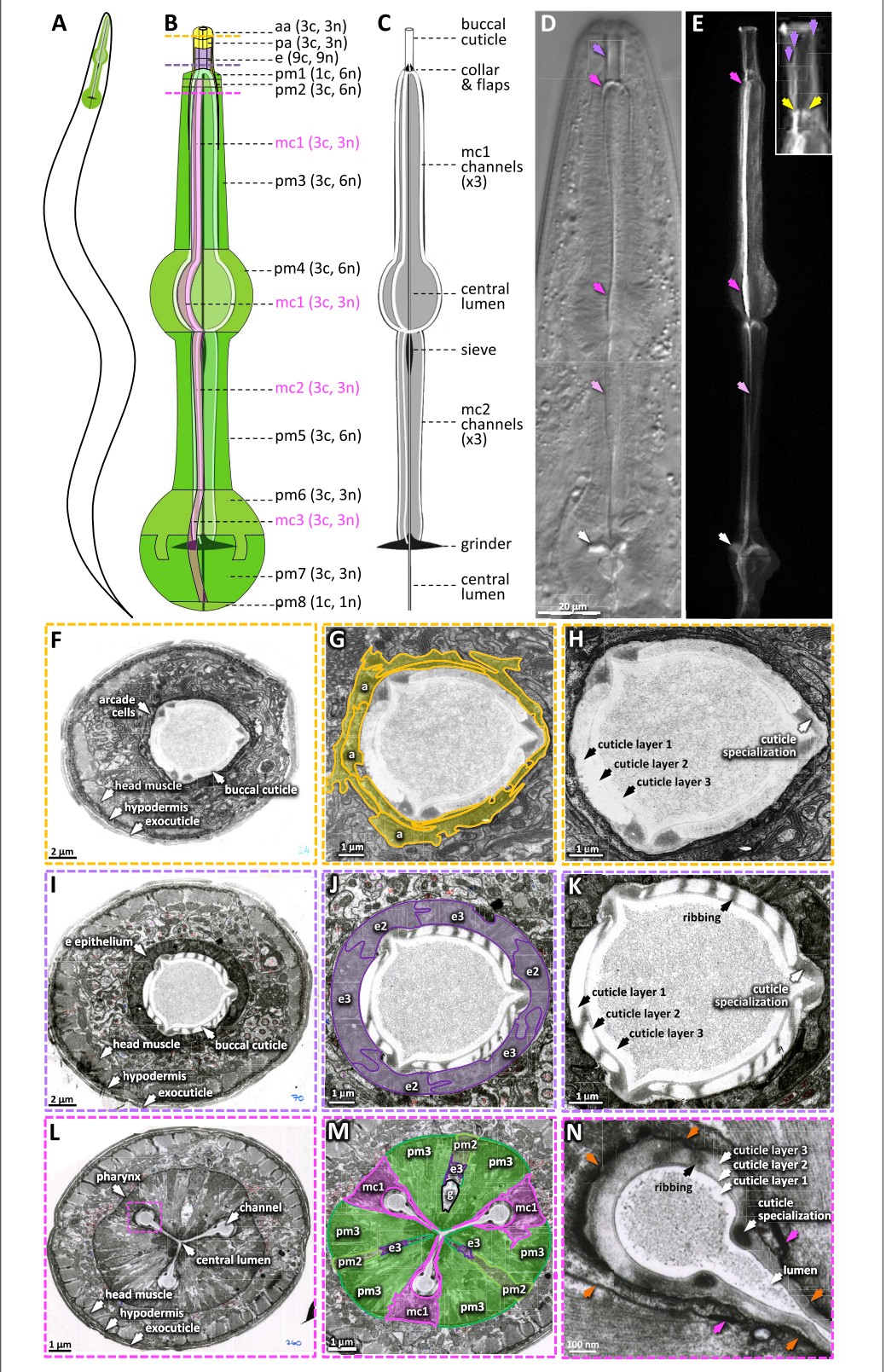

**Figure 1.** The pharyngeal cuticle and surrounding cells. (**A**) A schematic of the relative position of the *C. elegans* pharynx (green). (**B**) A schematic of the pharynx. The image of the outer cells is transparent, revealing the pharyngeal cuticle underneath. The 5 cells of the gland and the 20 pharynx-associated neurons are not shown. Each of the cell types are labeled followed by individual number of cells (**c**) and nuclei (**n**). aa, anterior arcade

*Figure 1 continued on next page*

*Figure 1 continued*

cells; pa, posterior arcade cells; e, pharynx epithelium; pm1-8, pharynx muscle; mc1-3, marginal cells. The yellow, purple, and pink dashed lines represent the area of the cross sections in (**F–H**), (**I–K**), and (**L–N**), respectively. (**C**) A schematic of the pharyngeal cuticle. Black and gray is cuticle; white is the lumen of the buccal cavity, central lumen, and channels. (**D, E**). Micrographs of the head of young adults expressing ABU-14::sfGFP. Differential interference contrast (DIC) is on the left and GFP of a similarly staged animal, taken with confocal microscopy, is on the right. Purple arrows show the buccal cuticle. The three purple arrows in the inset mark regions of ABU-14::GFP enrichment that likely correspond to the cuticle specializations noted in (**H**) and (**K**). Yellow arrows, the metastomal flaps; dark pink arrows, mc1 channel; light pink arrows, mc2 channel; white arrows, grinder. (**F–N**) TEM images taken from the *White et al., 1986* N2T series, stored on the WormAtlas EM archives. (**F–H**) show a cross section of the anterior buccal cavity; the surrounding arcade cells are highlighted in yellow in (**G**). (**I–K**) show a cross section of the posterior buccal cavity; the surrounding e epithelial cells are highlighted in purple in (**J**). (**L–N**) show a cross section of the procorpus posterior to the buccal cavity. In (**M**), the mc1 marginal cells associated with the channels are highlighted in pink; the pharyngeal muscles pm2 and pm3 are highlighted in green and 'g' indicates the gland. The pink box in (**L**) indicates the magnified area in (**N**). Orange arrows in (**N**) indicate the pm3-mc1 plasma membrane interface; the pink arrows indicate the adherens junctions.

---

with the expanded CFW and EY signal (orange arrows in *Figure 3A and E*). Second, CFW and EY brightly stain a prominent collar at the base of the buccal cavity (green arrows in *Figure 3A and E*). The amyloid-binding dyes stain the collar less (*Figure 3B and C*), and ABU-14::GFP fails to mark the collar (*Figure 3D*). In the TEM images, this collar is composed of light material. Hence, the electron-light material is likely enriched with chitin.

The CR dye and the ABU-14::GFP localize to the cuticle flaps (yellow arrows in *Figures 1E and 3B and D*), which are composed of the darker electron-dense material in the TEM (*Figure 3E*). The dark material of the flaps is contiguous with the dark ribbing of the buccal cuticle and the luminal-facing coating of the cuticle, all of which encapsulate the less electron-dense material (*Figure 3E*). An analogous organization is present in the cuticle that lines the channels (*Figure 3E*). Together, these observations suggest that the electron-dense material may be enriched in amyloid-like proteins and establish CR, ThS, CFW, and EY as useful markers of pharyngeal cuticle structure.

## Mining expression datasets yields a spatiotemporal map of pharyngeal cuticle development

To better understand pharynx cuticle construction, we built a spatiotemporal map of cuticle-centric gene expression by combining four published datasets (see *Figure 4—source data 1*). First, we anchored the map using a dataset that tracked gene expression levels in synchronized animals every hour for 16 hr from the mid L3-stage to adulthood at 25°C (*Hendriks et al., 2014*). This study identified 2718 genes whose expression oscillates during larval development with a peak in expression every 8 hr (p<0.001); this period corresponds to the 8 hr duration of the third and fourth larval stages at 25°C. Two of these 2718 genes have been retired due to reannotation. The 2716 genes can be grouped into bins of genes that peak at different larval development phases. For example, some genes peak during the first and ninth hour, others peak during second and tenth hour etc., such that there are successive waves of genes that oscillate through time (see Figure 1e of *Hendriks et al., 2014*). We present the 2716 genes from this dataset in the temporal order in which the genes peak in their expression over the 8 hr cycle (*Figure 4A*). We note that since we initiated our study an additional temporally resolved dataset has been published (*Meeuse et al., 2020*).

Second, we defined the interval on the map that corresponds to the molt by overlaying a dataset of genes that are upregulated during the L4 molt (p<0.001) (*George-Raizen et al., 2014*). The overlay indicates that molting peaks in the sixth hour on the map (*Figure 4A and B,, Figure 4—source data 1*). The fact that the genes that are upregulated during the L4 molt are clustered on the map provides reciprocal validation for both datasets (*George-Raizen et al., 2014*; *Hendriks et al., 2014*). We herein routinely refer to hour 6 as the reference peak molting hour.

Third, we identified the genes on the temporal map whose expression is enriched in the cells surrounding the pharyngeal cuticle relative to all other tissues. We did this by overlaying single-cell expression data from cells isolated from L2-staged animals (*Cao et al., 2017*). We found 367 'pharynx'-enriched transcripts (≥1.5-fold enriched in the pharynx relative to all other tissues and at least 25 transcripts per 1 million reads) that oscillate over time (*Figure 4A, Figure 4—figure supplement 1,*

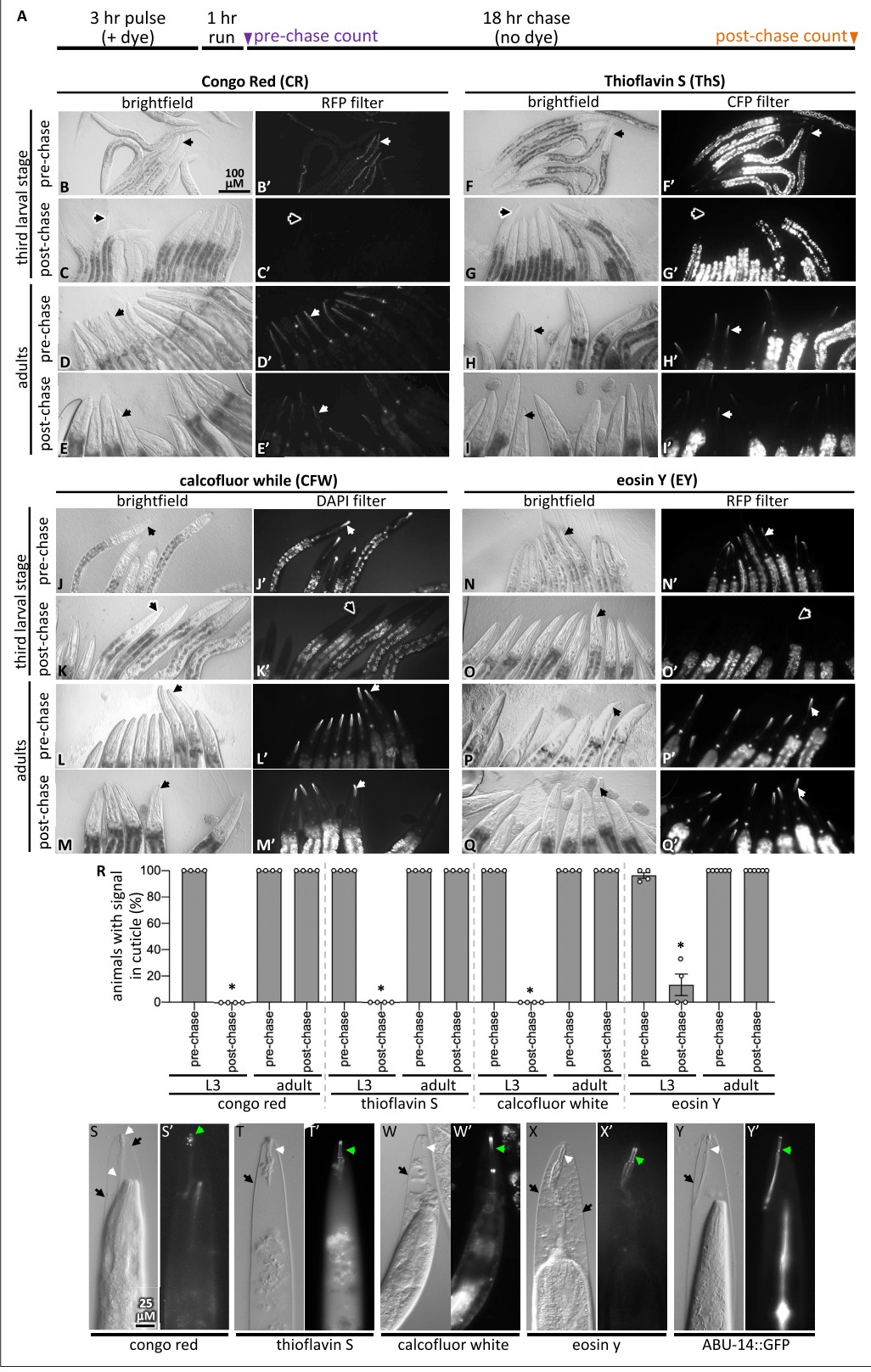

**Figure 2.** Pulse-chase and cuticle mutant experiments show dye association with the cuticle. (**A**) Schematic showing the pulse-chase assay. Synchronized populations of L3 or adult worms were incubated with a dye for 3 hr (the 'pulse'), after which worms were washed with M9 and run on normal plates with food for 1 hr. Worms are transferred to fresh plates and the presence of the dye was scored (see 'Materials and methods' for details).

*Figure 2 continued on next page*

*Figure 2 continued*

Then, 18 hr later (i.e., after the chase), worms were again scored for the presence of the dye. (**B–Q**) In each of the four groups of eight micrographs with the dye indicated in the header, the top two rows show the pulse-chase experiment done starting with L3s, and the bottom two rows show the pulse-chase experiment done with adults. The filter used to visualize the dyes is indicated at the header of the rightmost column in each of the four panel sets. In all panels, white arrows highlight the presence of the dye in the cuticle and black arrows show cuticle without dye signal. The scale bar is indicated. (**R**) The fraction of worms with stained cuticle before and after the chase for each dye is shown; a minimum of four repeats (**N**) were done with a sample size of 7–34 animals (average = 13) (**n**) per repeat. Asterisk denotes statistically significant difference relative to the pre-chase values (p<0.05). Standard error of the mean is shown. (**S–X**) Wildtype animals treated with *mlt-9(RNAi)* that are incubated with the indicated dye for 3 hr. The brightfield differential interference contrast (DIC) image and the corresponding fluorescent image are shown for each treatment. (**Y**) An animal expressing transgenic ABU-14::GFP treated with *mlt-9(RNAi)* but without dye stain. The scale in (**S**) applies to all panels.

The online version of this article includes the following figure supplement(s) for figure 2:

**Figure supplement 1.** The fluorescence and filter controls for dye staining.

*Figure 4—source data 1*). This set of genes includes those enriched in expression within the pharyngeal epithelium, muscles, and gland cells, but not pharyngeal-associated neurons.

Fourth, we determined the likelihood of gene products being secreted using Signal P (v4.1) predictions extracted from the WormBase Parasite database to identify signal peptides (with scores of 0.45 or more) genome-wide (*Hertz-Fowler and Hall, 2004*). We recognize that while this approach is systematic, Signal P does not identify all secreted or plasma membrane-associated transmembrane proteins. The oscillating pharynx-enriched set contained 226 genes (62%) that encode a signal peptide (*Figure 4A*, *Figure 4—source data 1*). By comparison, only 39% of the remaining oscillating gene set (n = 2349) and only 17% of the entire non-oscillating genes of the genome (n = 17,614) encode a signal peptide (*Figure 4—source data 1*). The temporal map shows a concentration of genes that peak in expression from the pharynx and are secreted at the time of molting (*Figure 4A*).

We investigated the change in transcript abundance in the pharynx over the cyclical 8 hr window of larval development for the oscillating genes. We found a nearly 30-fold increase in transcript abundance for those gene products predicted to be secreted relative to the global average of pharynx gene expression during the peak molting hour (*Figure 4B*). There is a shoulder of peak expression at hour 7 for those non-secreted gene products (*Figure 4B*) that may correspond to the increase in tissue growth after the molt. *Cao et al., 2017* further dissected their single-cell sequencing data into tissue subtypes. We find that the expression of predicted secreted products from the pharynx epithelial cells peaks dramatically during the peak molting hour, whereas pharynx gland transcription peaks in the preceding hour (*Figure 4*). Non-secreted epithelial and muscle products peak in expression during hour 7 (*Figure 4C and E*). Given that mRNA expression levels are positively correlated with protein abundance in invertebrate systems (*Ho et al., 2018*; *Schrimpf et al., 2009*), we conclude that there is a likely a burst of proteins secreted in preparation for the molt.

## Orthogonal data validate the spatiotemporal map

We explored the validity of the spatiotemporal map in four ways. First, previous work established that the molting of the body cuticle precedes that of the pharyngeal cuticle (*Wright and Thomson, 1981*). We therefore expected a peak in gene expression from the hypodermis that precedes that of the pharynx, which is what we observe (*Figure 5A*).

Second, we systematically investigated published reports of expression (not including the datasets used to build the spatiotemporal map) for the 226 oscillating pharynx secretome genes. In this analysis, we also included the 17 additional genes of special interest called out in *Figure 4A* that include *myo-1*, *myo-2*, and *myo-5* for example (see *Supplementary file 1* for details). We surveyed Yuji Kohara's whole-mount RNA in situ database (*Motohashi et al., 2006*) and literature reports of transgene and sequencing-based expression patterns curated by WormBase to determine whether there is additional evidence that these 243 genes are enriched in expression within the pharynx (*Supplementary file 1*). 83 (34%) of the 243 genes lacked reported expression patterns in the Kohara and WormBase databases. Of the remaining 160, 152 (95%) demonstrate a clear enrichment of expression within the pharynx (*Figure 5B*; *Supplementary file 1*).

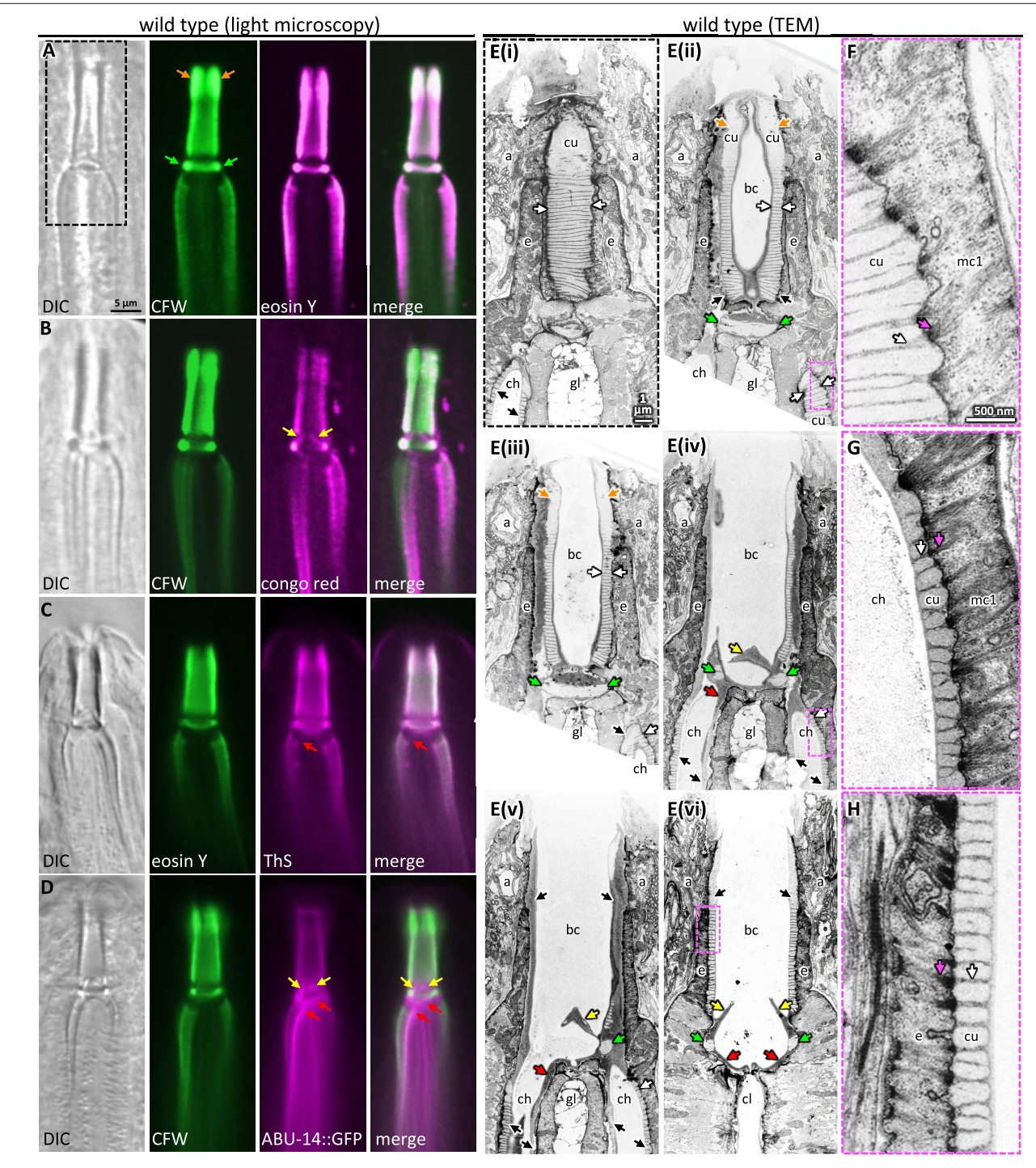

**Figure 3.** Probing pharyngeal cuticle composition with characterized dyes. (**A–D**) Images of the buccal and mc1 channel cuticles and surrounding cells. The dyes or GFP-fusion protein examined is indicated. DIC, differential interference contrast; CFW, calcofluor white; ThS, thioflavin S. The scale shown in (**A**) is the same for (**B–D**). (**E**) Serial coronal sections of unpublished transmission electron micrographs taken by *Wright and Thomson, 1981*. The scale in (**E(i)**) applies to all images in the E series. (**F–H**) Magnifications of the boxed areas highlighted in the images to the left. (**G**) represents a slightly different plane than that depicted in (**E(iv)**) and was chosen because of the clearly visible filaments. The scale in (**F**) is the same as that for (**G**) and (**H**). For all panels: a, arcade cells; ch, mc1 channel; cu, cuticle; e, e epithelium; gl, gland cell; bc, buccal cavity; cl central lumen; orange arrows, the anterior enlargement of the buccal cuticle; green arrows, the prominent ring at the base of the buccal cavity; yellow arrows, the electron-dense flaps at the base of the buccal cavity; red arrows, the electron-dense material at the anterior end of the channel cuticle; black arrows, pharyngeal cuticle when too small to be labeled with 'cu'; white arrows, the ribbing of the pharyngeal cuticle; pink arrows, the cytoplasmic filaments that correspond to the abutment of the ribbing.

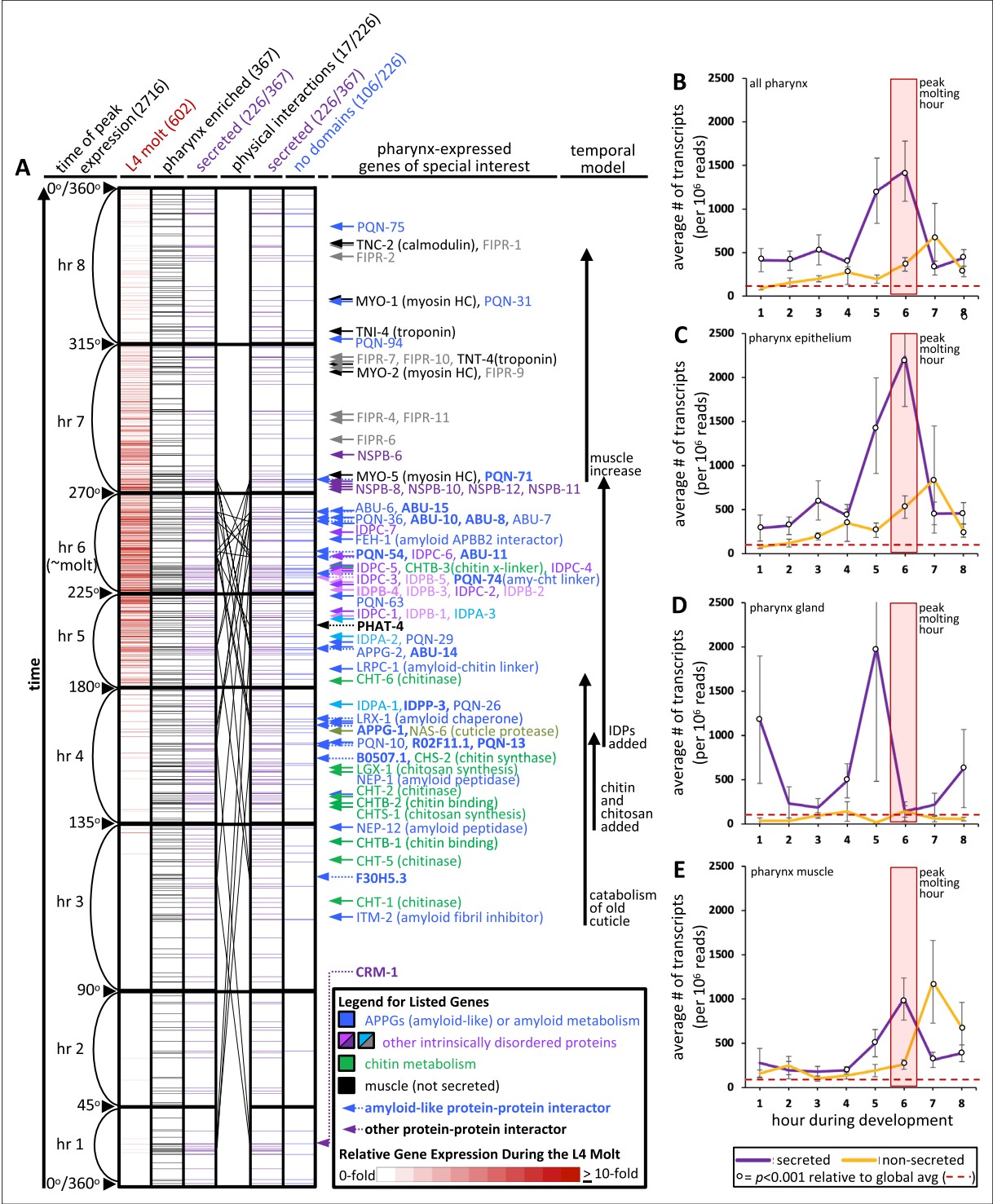

**Figure 4.** An informatic reconstruction of the pharyngeal cuticle. (**A**) A chart of 2716 genes whose expression oscillates over larval development with a periodicity that corresponds to larval stages. See text for details. Each row represents a single gene. Rows are arranged along the y-axis in order of the time at which each gene reaches its peak expression level with those earliest in the time course at the bottom and those latest in the period at the top. Because the periodicity is a continuum during larval development, *Hendriks et al., 2014* represented time as degrees of a circle. That concept is preserved here, and the degree is indicated along the y-axis and divided into bins of time relative to the molting period. The first data column (red) represents the 602 oscillating genes that also were found to be upregulated in expression during the L4 lethargus (molting) period (see Supplemental Table 1 in *George-Raizen et al., 2014*); the scale of the relative expression level from this independent study (*George-Raizen et al., 2014*) (is indicated in the legend). The second data column (black) represents the 367 genes from the set of 2716 that are enriched in expression in the pharynx (data from

*Figure 4 continued on next page*

*Figure 4 continued*

*Cao et al., 2017*; see *Figure 4—source data 1* for enrichment). The purple columns show the 226 genes (of the 367 pharynx-enriched set) that are predicted to be secreted. They are duplicated to show the 26 protein–protein interactions (PPI) among the 17 oscillating pharynx-secreted proteins identified through Genemania (see *Figure 5C*, *Figure 4—source data 1*, and *Figure 5—source data 1* for the details of which protein pairs interact). The identity of the interacting proteins is indicated with bold lettering and a dotted arrow on the right of the graph. The last column (blue) represents those pharynx-enriched genes that lack an obvious domain as predicted by WormBase, PFAM, and SMART databases (see text for details). 78 pharynx-expressed genes of special interest are indicated with arrows to the right of the graph. The color of the arrows and text corresponds to broad categories indicated in the legend. (**B–E**) The average number of transcripts produced by genes whose expression is enriched in the indicated tissue as a function of developmental time. In all graphs, results are binned according to the hours indicated in (**A**), the global average transcript number (49.33) is indicated by the red dotted line. Statistical differences were measured using a Student's *t*-test against the global average of gene expression levels in the pharynx. Standard error of the mean is shown in all graphs. The peak molting hour in (**B–E**) is highlighted by the transparent red box.

The online version of this article includes the following source data and figure supplement(s) for figure 4:

**Source data 1.** This is the master file with all relevant data for the spatiotemporal map.

**Figure supplement 1.** Tissue-enriched expression levels of tissue-enriched classes of genes.

Third, we reasoned that the pharynx secretome might be rich in protein–protein interactions (PPIs) because many of the secreted proteins likely interact to form a matrix. We explored PPIs systematically using Genemania, which is an online tool that facilitates the analysis of experimentally derived interaction data curated from the literature (*Franz et al., 2018*). To analyze each tissue's secretome, we returned to the *Cao et al., 2017* single-cell sequence datato parse the proteome into proteins that are enriched in the major tissues using the same criteria described above for the pharynx (*Figure 4—source data 1*). These tissues included the pharynx (470 proteins), body wall muscles (BWMs) (326 proteins), glia (426 proteins), gonad (832 proteins), hypodermis (411 proteins), intestine (781 proteins), and neurons (965 proteins) (*Figure 4—figure supplement 1*). We separated out the 166 collagens from the proteome because of their unique sequence properties. The remaining 15,892 proteins are binned into a non-specific group. For each of these groups, we parsed them into those encoding a signal peptide, and those without. Genemania reports multiple lines of evidence for 36 PPIs among a network of 20 proteins within the pharynx secretome (*Figure 5C*). This interaction network is denser than that from most other secretomes (*Figure 5—figure supplement 1*, *Figure 5—source data 1*).

Fourth, literature searches reveal that the spatiotemporal map includes many genes with known roles in pharynx development (*feh-1*, *myo-1*, *myo-2*, *nep-1*, *pqn-75*, *sms-5*, *tnc-2*, and *tni-4*) and the few genes known to play roles in pharynx cuticle formation (*abu-6*, *abu-14*, *chs-2*, and *nas-6*) (*Supplementary file 1*). We further investigated the functional relevance of the map by conducting a survey of publicly available mutants of genes predicted to contribute to the pharyngeal cuticle. Light microscopy revealed obvious cuticle defects in the pharynx of animals harboring disruptions of *feh-1*, *idpa-3*, *idpc-1*, *lrpc-1*, and the positive control *nas-6* (*Figure 5D*; *Supplementary file 1*), bringing the total number of genes with known pharynx cuticle defects to 7 of the 243 genes listed in *Supplementary file 1*. The pattern of amyloid and chitin dyes is unaligned in the *feh-1*, *idpa-3*, *idpc-1*, and *lrpc-1* mutants (*Figure 5D*). This not only provides insight into the proteins' importance in cuticle structure, but reinforces the idea that the two dyes recognize distinct components within the cuticle.

Finally, we further confirmed the map's ability to predict spatial expression patterns by inserting green fluorescent protein coding sequence in frame with five poorly characterized gene products, namely, IDPA-3, IDPB-3, IDPC-1, FIPR-4, and NSPB-12 (*Figure 6*). We also included the previously characterized ABU-14::GFP (*Figure 6A*). We counterstained the resulting transgenic animals with CFW to interrogate the spatial overlap of the tagged proteins with the chitinous cuticle. As predicted, we found that all five reporters are expressed exclusively in association with the pharynx and overlap in their localization with the pharynx cuticle. Briefly, tagged IDPA-3 was enriched in the grinder, overlapping the CFW-stained component and lining of the terminal bulb cuticle. In addition, we observed enrichment of tagged IDPA-3 in the presumptive ECM that lies between the terminal bulb and the intestinal valve (white arrow in *Figure 6B*). Tagged IDPB-3 was expressed weakly and localized exclusively to the pm6 cells and material surrounding the CFW-stained grinder (*Figure 6C*). Tagged IDPC-1 had a similar pattern to that of tagged ABU-14; associating with both the anterior and posterior components of the pharyngeal cuticle. However, tagged ABU-14 appears to localize adjacent to CFW-stained components whereas tagged IDPC-1 overlaps CFW-stained components (*Figure 6A and D*, *Figure 6—figure supplement 1*). Tagged NSPB-12 localized to the anterior pharynx cuticle

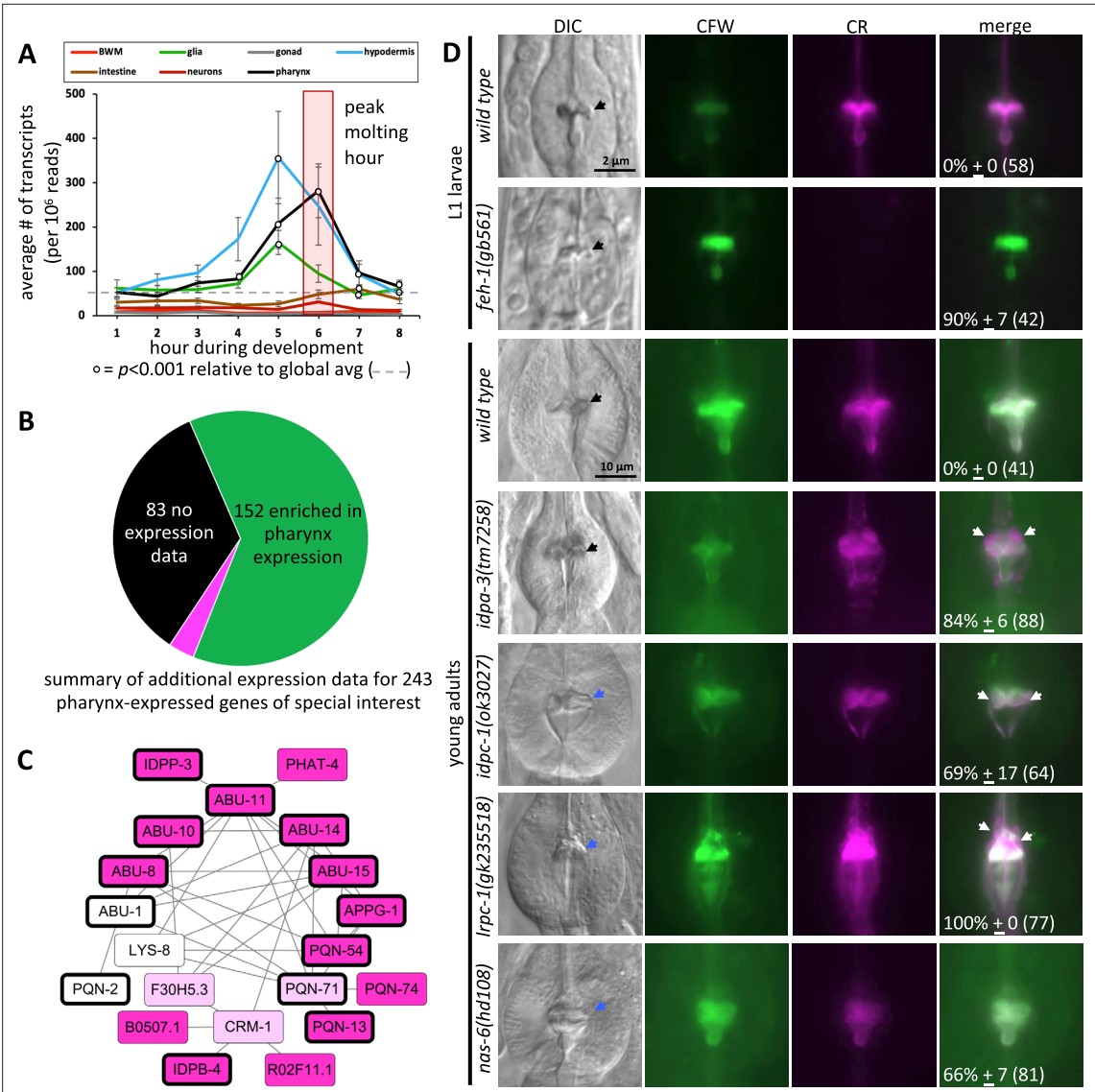

**Figure 5.** The spatiotemporal map has predictive power. (**A**) Average gene expression in each of the indicated tissue types plotted as a function of developmental time. In the first hour of the time course, for example, 219 genes peak in expression and the average expression of each of these 219 genes in each of the indicated tissues is plotted for hour 1 on the graph. Standard error of the mean is shown. The peak pharynx molting hour is highlighted by the transparent red box. Significant differences relative to the global mean is calculated with a Student's *t*-test. (**B**) A pie chart summarizing the search of publicly available information on previously documented expression patterns of the 226 oscillating pharynx secretome genes and 17 other genes of interest (which are part of the 78 genes highlighted in *Figure 4A*). Published expression patterns could be found for 160 of the 243 genes. Of the 160, the expression pattern of only 8 genes (indicated in fuchsia) did not support clear enrichment in the pharynx. See *Supplementary file 1* for details. (**C**) Protein–protein interactions within the pharynx secretome. Dark pink nodes are those genes that peak in expression during hours 4, 5, or 6 on the spatiotemporal map. Light pink nodes peak in expression outside of hours 4, 5, or 6. White nodes represent genes that do not oscillate. Nodes outlined in bold are those proteins composed of >75% intrinsically disordered regions (IDRs). (**D**) A survey of mutants for obvious pharynx cuticle defects. Each of the indicated backgrounds are stained with calcofluor white (CFW) and Congo Red (CR). The mean percentage of animals showing defects, together with the standard error of the mean (N = 3 independent trials with more than eight animals each trial). The total number of animals surveyed is indicated in brackets. The scale for L1 and adult animals is shown. DIC, differential interference contrast, black arrows indicate a normal terminal bulb grinder, blue arrows indicate a dysmorphic grinder, and white arrows indicate discordant CR staining.

The online version of this article includes the following source data and figure supplement(s) for figure 5:

**Source data 1.** Supporting information for the network diagram in *Figure 5C* and related insights.

**Figure supplement 1.** The pharynx secretome has a dense protein–protein interaction (PPIs) network relative to other secretomes.

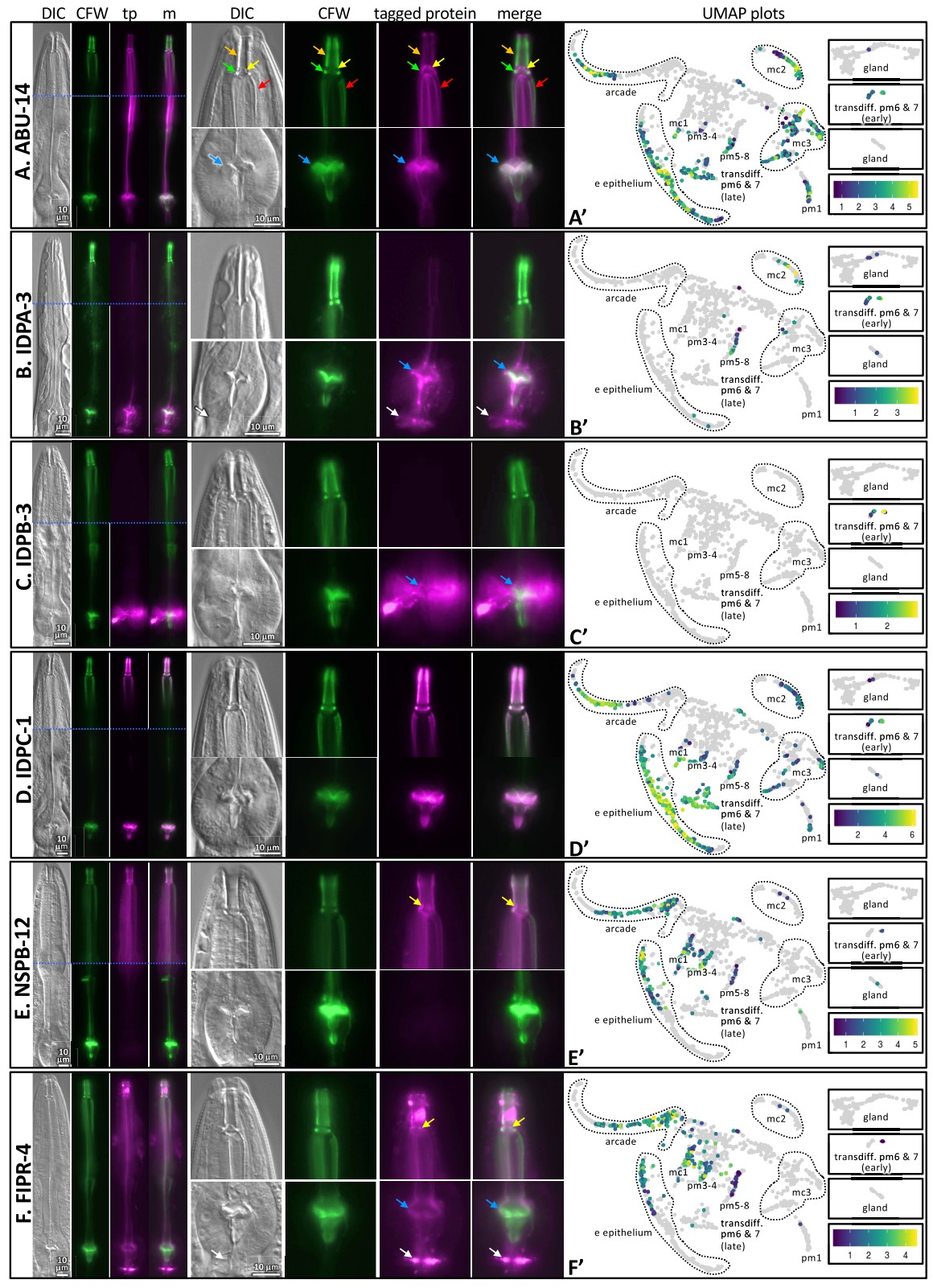

**Figure 6.** The localization of six fluorescently tagged pharynx cuticle components. Each of the six large horizontal boxes contain data about the six predicted gene products indicated on the left. In each box, the four images on the left are of the head of a single worm, imaged first with differential interference contrast (DIC), then with calcofluor white (CFW) in green, then the fluorescently tagged protein (tp) protein of interest in fuchsia, followed by a merged (m) image as indicated at the top of the columns. A blue horizontal line indicates the intersection of two cropped images to show

*Figure 6 continued on next page*

*Figure 6 continued*

different relevant focal planes of the same animal. The scale is indicated. The middle set of eight images correspond to magnified buccal cavity and channels (top) and terminal bulb and grinder (bottom). The scale is indicated. Colored arrows are used for reference in (**A**) and used to draw attention to particular features in (**B–F**): Orange, buccal cavity; yellow, flaps; green, collar; red, anterior channels; blue, grinder; white, presumptive ECM between the terminal bulb and intestinal valve. The graph on the right is a UMAP plot of the pharynx mRNA expression pattern for the respective gene (see text for details). The relative expression level is indicated. (**A**) An RP3439 animal harboring the *trIs113[Pabu-14:abu-14:superfolderGFP; rol-6(d); unc-119(+)]* integrated array. (**B**) An RP3519 animal harboring the *Ex[idpa-3p::IDPA-3::mNeonGreen; myo-2p::mCherry]* extrachromosomal array. (**C**) An RP3498 animal harboring the *Ex[idpb-3p::IDPB-3::mNeonGreen; myo-2p::mCherry]* extrachromosomal array. (**D**) An RP3497 animal with genomic *idpc-1* fused in-frame to the coding sequence for mGreenLantern. (**E**) An RP3499 animal harboring the *Ex[nspb-12p::NSPB-12::mNeonGreen; myo-2p::mCherry]* extrachromosomal array. (**F**) An RP3514 animal harboring the *Ex[fipr-4p::FIPR-4::mNeonGreen; myo-2p::mCherry]* extrachromosomal array. All animals are counterstained with the calcofluor white (CFW) chitin stain. The expression patterns shown are typical of the population that are positive for the transgene.

The online version of this article includes the following figure supplement(s) for figure 6:

**Figure supplement 1.** A comparison of the tagged ABU-14 and IDPC-1 localization patterns.

components exclusively, including that of the buccal cavity, flaps, and anterior channels (**Figure 6E**). Tagged FIPR-4 localized to both anterior and posterior pharynx cuticle components (but not the grinder teeth proper) and the presumptive pharynx-intestinal valve ECM (**Figure 6F**). Together, these analyses provide confidence in the predictive value of the spatiotemporal map.

## The pharynx secretome is enriched in proteins with high predictions of phase separation

To better understand the types of proteins that are secreted by the pharynx, we manually curated the domain organization of all 367 oscillating pharynx-enriched gene products as reported by the WormBase, SMART, and PFAM protein databases (**Letunic and Bork, 2018**; **El-Gebali et al., 2019**; **Figure 4—source data 1**). We found that 106 of the 226 secreted proteins (47%) lacked any defined domain (last column of the chart in **Figure 4A**, **Figure 4—source data 1**). This prompted a systematic investigation of low-complexity sequence within the pharynx secretome using NCBI's SEG algorithm (**Wootton and Federhen, 1993**). Indeed, we found the pharynx secretome to be greatly enriched with low-complexity regions (LCRs) (p=1E-69) (**Figure 7A**). Given that low complexity is tightly associated with intrinsic disorder, we used the Spot-Disorder algorithm (**Hanson et al., 2017**) to systematically analyze whether the pharynx secretome is also enriched for IDRs and found that it is (p=8E-10) (**Figure 7B**).

Low-complexity intrinsically disordered protein regions often provide multivalency that can enable a protein to transition from being soluble to becoming a phase-separated liquid, gel, stable polymeric matrix, or an insoluble amyloid (**Muiznieks et al., 2018**). We explored the potential of the different protein sets to phase separate using three different predictive algorithms, including PSPredictor (**Chu et al., 2022**), PLAAC (**Lancaster et al., 2014**), and LLPhyScore (**Cai et al., 2022**). PLAAC was originally designed to scan for prion-like sequences, but has been retrospectively used as a reliable tool to predict phase separation (**Vernon and Forman-Kay, 2019**). Each algorithm reveals that the pharynx secretome is enriched in proteins with phase separation capability (p=2E-46, p=2E-52, and p=2E-31, respectively) (**Figure 7C–E**).

We also examined low-complexity, intrinsic disorder and phase-separation propensity as a function of developmental time. The peak molting hour corresponds to a clear peak in low-complexity and intrinsic disorder of secreted products (**Figure 7A' and B'**). The other three predictors also show significant peaks in phase separation propensity of secreted products during the peak molting hour, but variably show peaks at other time points as well (**Figure 7C'–E'**). To better understand the relative abundance of gene products with the specific sequence features highlighted in **Figure 7A'–E'**, we multiplied the trait value for each gene with the relative number of transcripts for each respective gene. In this light, we see a striking peak of all trends at the peak molting hour (**Figure 7A"–E"**). This analysis suggests that the pharyngeal cuticle is likely flooded with low-complexity, intrinsically disordered proteins with phase separation potential during the peak molting hour.

Finally, we tested these predictions by asking whether IDPC-2 can phase separate. Upon cleaving off the MBP affinity tag from the in vitro-expressed proteins, we see that IDPC-2 and the positive control FUS can form phase-separated droplets (**Figure 8A and B**). In these experiments, we use a

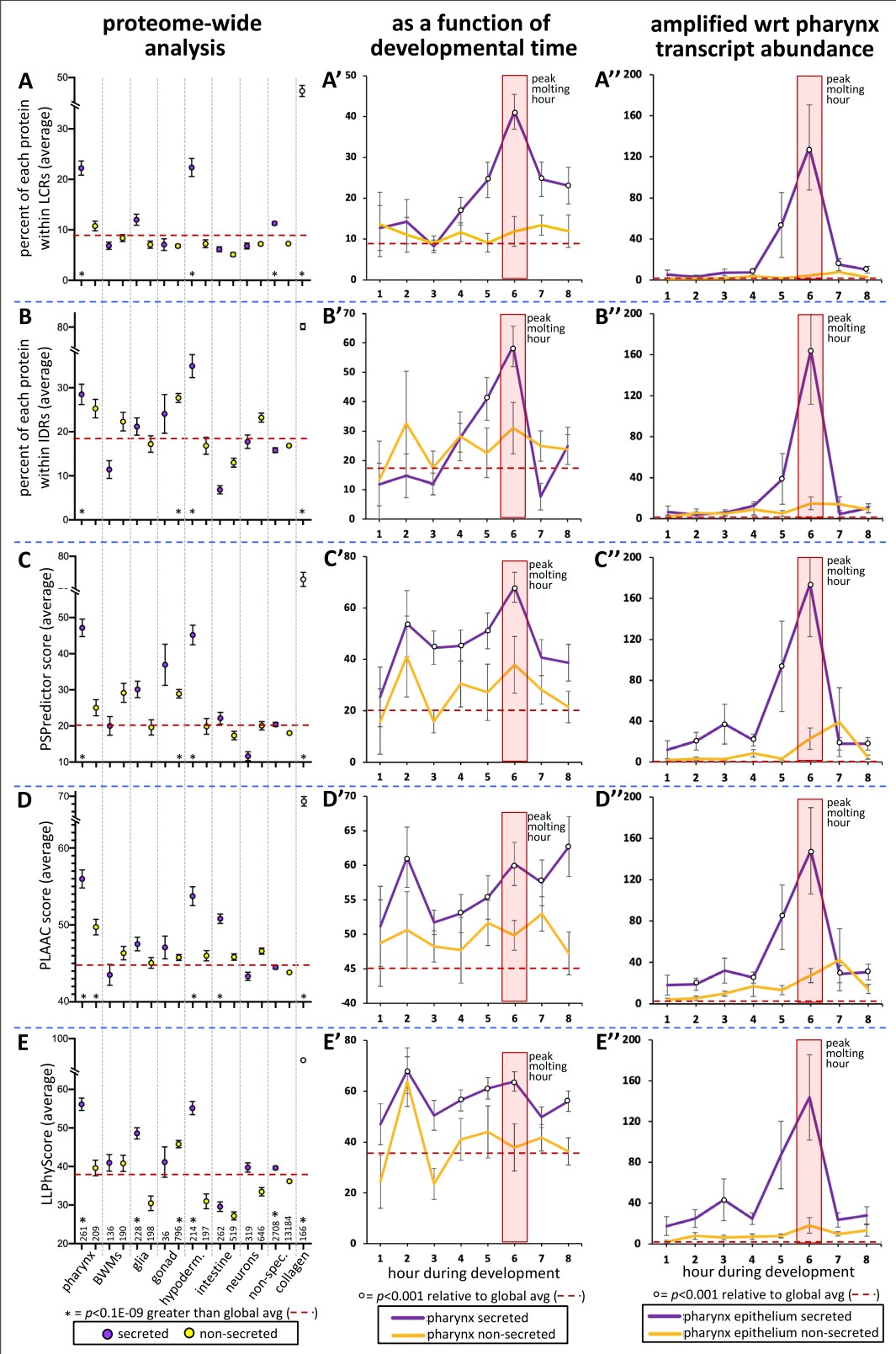

**Figure 7.** The pharynx secretome is enriched with intrinsically disordered proteins with phase separation capability. (**A–E**) An analysis of the entire proteome for the indicated properties. The tissue type examined, as well as the number of genes in each bin, is indicated at the bottom of the graph in (**E**) (hypoderm., hypodermis; non-spec., non-specific). Statistical differences, indicated with an asterisk at the bottom of each graph, were measured

*Figure 7 continued on next page*

*Figure 7 continued*

using a Student's *t*-test against the global average (indicated with a red hatched line for each property). (**A'–E'**) An examination of the same properties as (**A–E**), but with a focus on genes whose expression is enriched in the pharynx over developmental time. (**A''–E''**) An examination of the same properties as (**A'–E'**), but normalized with respect to each gene's transcript abundance within the pharyngeal epithelium. For each gene, the number of transcripts was multiplied by the value of gene products property (i.e., % within low-complexity region [LCR], % withing intrinsically disordered regions [IDRs], or PSPredictor score, etc.), and the average for that temporal bin was calculated. The Y-axis in (**A''–E''**) reports numbers in the thousands. Statistical differences were measured using a Student's *t*-test against the global average. In all graphs, standard error of the mean is shown. Because the PLAAC algorithm can report negative scores up to –60, 60 was added to the PLAAC scores of all gene products for the sake of clarity. The peak molting hour is highlighted by the transparent red box.

molecular crowding reagent (Ficoll) to mimic in vivo molecular crowding (*André and Spruijt, 2020*). These data support the informatic analyses that predict that many of the proteins incorporated into the cuticle may be capable of phase separation.

## The pharynx secretome is not enriched with amyloidogenic proteins

We investigated the propensity of pharynx secretome proteins to form filaments. We first used the LARKS algorithm that predicts kinked b-structure, which can drive proto-filament assembly and reversible fiber formation (*Hughes et al., 2018*). Indeed, we find a significant enrichment in LARKS scores within the pharynx secretome (*Figure 8C*). This prediction is corroborated by the LLPhyScore predictor of kinked b-structure (*Figure 8D*). We also investigated whether the pharynx secretome is enriched in amyloidogenic proteins. Both the Budapest (*Keresztes et al., 2021*) and AmyloGram (*Burdukiewicz et al., 2017*) machine-learning predictors, as well as the structure-based PATH predictor (*Wojciechowski and Kotulska, 2020*), fail to show any enrichment within the pharynx secretome of amyloidogenic proteins (*Figure 8E–G*).

We further probed the ability of the pharynx secretome to form amyloid fibers using CR dye. CR has long been used as a diagnostic tool to identify rigid amyloid fibrils because of its special property of emitting apple green birefringence upon binding the ordered fibril array in the presence of polarized white light (*Divry, M, 1927*). This is in sharp contrast to the colorless birefringence of the crystalizing compounds (*Figure 8H*). While CR specifically stains the pharynx cuticle, we found that CR-stained cuticles do not emit apple green birefringence (n > 30) (*Figure 8I and J*). We are confident that our imaging system is capable of detecting CR-derived apple green birefringence because of a serendipitous observation. We found that when CR is co-incubated with a small molecule (called wact-190) that forms crystals in the pharyngeal cuticle (*Kamal et al., 2019*), the resulting crystals exhibit apple green birefringence (*Figure 8K*). We infer that this happens because CR likely becomes incorporated into a regular array, that is, the wact-190 crystal. Together, these results indicate that it is unlikely that the cuticle harbors rigid amyloid fibrils, which is consistent with both the flexible nature of the pharynx cuticle (*Huang et al., 2008*; *Avery, 1993*) and the absence of any detectable amyloid-like fibers in previous transmission electron micrographs of the pharynx cuticle (*Wright and Thomson, 1981*; *White et al., 1986*). We conclude that the pharynx secretome is likely enriched in proteins with intrinsic disorder, phase separation capability, and proto-filament formation capability, but not enriched with proteins that form rigid amyloid fibrils.

## The transcripts encoding secreted IDR protein families peak in expression in overlapping waves during cuticle construction

Given the enrichment in low-complexity sequence within the pharynx secretome, we were curious to know whether it has any global bias in amino acid residue distribution relative to other protein sets. We found a significant enrichment of nine residues with a strong bias against charged and hydrophobic residues (at least p<2E-05; *Figure 9A*). Upon considering relative abundance of amino acid residues as a function of time, we see that proteins rich in cysteine, proline, and glutamine peak in expression during new cuticle construction (*Figure 9B*).

We used the Clustal Omega clustering tool (*Sievers et al., 2011*) to determine whether there were families of proteins with similar sequence within the 106 proteins that lacked domains within the pharynx secretome. We found six distinct families of low-complexity proteins through this analysis

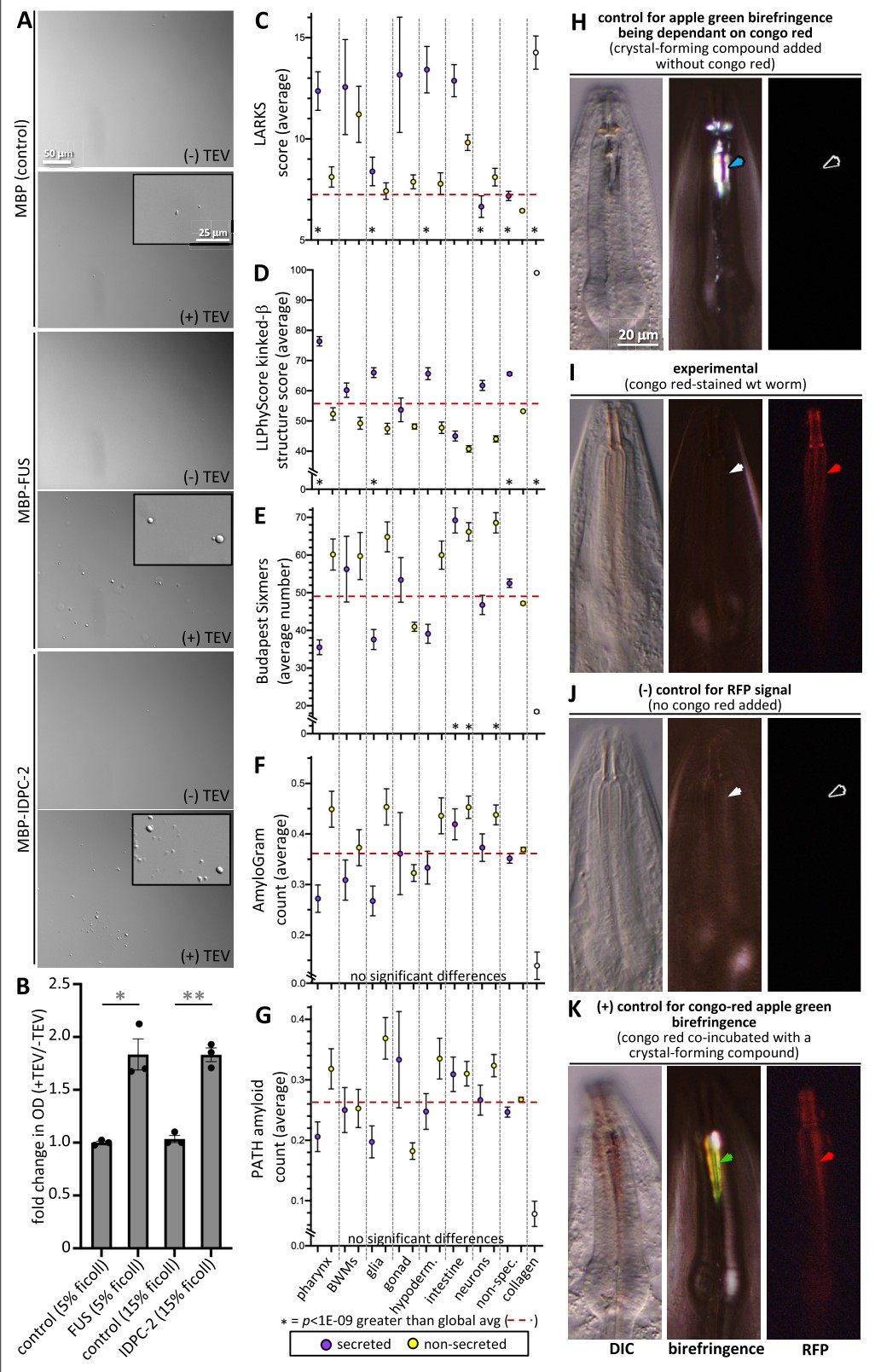

**Figure 8.** Cuticle proteins can likely phase separate and are enriched with protofilament but not amyloidogenic sequence. (**A**) In vitro purified maltose-binding protein (MBP) control (15% Ficoll) or fusions with the FUS-positive control (5% Ficoll) and IDPC-2 (15% Ficoll) phase separate into spheres upon cleaving off the MBP tag with TEV protease, while the MBP-only negative control does not. The inset is a magnification of the corresponding area.

*Figure 8 continued on next page*

*Figure 8 continued*

The scales for all insets and larger images are respectively the same. (**B**) Quantification of the fold change in optical density (OD; 395 nm) of indicated samples after 1 hr of treatment with TEV relative to the OD without the addition of TEV. *p<0.01 and **p<0.001, respectively, using a Student's *t*-test. In (**A, B**), all proteins are at a concentration of 1 mg/mL, except for FUS, which is at 1.5 mg/mL. The MBP-only control is therefore a vast molar excess. (**C–G**) An analysis of the entire proteome for the indicated properties. The details are the same as that indicated for *Figure 7*. (**H**) Control for the dependence of the apple green color on Congo Red (CR). Wildtype animals are incubated in wact-190 as previously described (*Kamal et al., 2019*) and yield birefringent crystals that lack notable apple green color (blue arrowhead). (**I**) Wildtype adult worms incubated with CR exhibit red fluorescent pharyngeal cuticle (red arrowhead; left column), but no apple green birefringence (white arrowhead; middle column). Differential interference contrast (DIC) is shown in the left column. Zero out of 30 animals exhibited apple green birefringence. (**J**) Control for the CR RFP signal. Wildtype animals are incubated without CR present. No birefringence (white arrowhead) or CR signal (black arrowhead) results. (**K**) Control for the ability to detect CR apple green birefringence. The wildtype animal was incubated simultaneously in CR and wact-190, a small molecule that crystalizes in the pharyngeal cuticle. The apple green birefringence (green arrowhead) manifests under these conditions because CR likely incorporates into the regular crystal lattice of the wact-190-derived crystals. The scale in (**H**) is representative of all panels.

(*Figure 9C*, *Figure 9—source data 1*). Members of each family share an enrichment of particular residues (*Figure 9D*), contain regions of high percentage positional sequence identity (*Figure 9E*, *Figure 9—figure supplement 1*), and are expressed at similar times as one another (*Figures 4A and 9F*). These six families include three new families of IDR-rich proteins, which we have named IDPA, IDPB, and IDPC, a subgroup of APPGs (*George-Raizen et al., 2014*; *Figure 9E*), and the relatively short NSPBs and FIPRs about which little is known. See *Supplementary file 1* for all newly named genes presented in this study and *Supplementary file 2* for all members of the six families described here. Systematic searches relying on positional alignment reveal no obvious homologs of these six families in any group beyond Nematoda (WormBase). Furthermore, a comparison of the consensus sequence from these families (*Figure 9E*, *Figure 9—figure supplement 1*) to the cuticle proteins of other Ecdysozoans (*Willis, 2010*) reveals no obvious similarity in the pattern or amino acid sequence biases.

The transcription of the six families of low-complexity proteins peaks in expression in successive overlapping waves, with five of the waves concentrated around the peak molting hour (*Figure 9F*). The combined use of the three different predictors of phase separation suggests that the IDPAs, IDPBs, IDPCs, and the APPGs may be able to phase separate (*Figure 9F*). The FIPRs and NSBPs are also likely to phase separate but fail to score high with the SpotDisorder algorithm because of their small size. The IDPAs and IDPBs are predicted to form protofilaments (as measured by LARKS), the IDPAs and APPGs score especially high with the prion sequence evaluator (PLAAC), and five members of the APPGs (ABU-6, ABU-7, ABU-8, ABU-15, and PQN-54) are predicted to be amyloidogenic (as measured by AmyloGram and PATH) (*Figure 9F*). These results further support the idea that a large proportion of the proteins secreted by the pharynx during cuticle construction are IDR-rich with phase-separating capability.

## Epithelial and transdifferentiated cells secrete abundant products during the molt

We sought increased spatial resolution of peak gene expression that is associated with pharyngeal cuticle construction over the course of the temporal map. We therefore returned to the Cao et al. single-cell sequencing dataset (*Cao et al., 2017*; *Packer et al., 2019*) to systematically visualize the expression patterns of pharynx secretome components. *Cao et al., 2017* and *Packer et al., 2019* identified 1675 sequenced cells that belong to the pharynx. When grouped according to similar expression profiles, the pharynx cells form subclusters on a Uniform Manifold Approximation and Projection (UMAP) created by Packer et al. (see https://cello.shinyapps.io/celegans_L2/) that represent cells of a similar type (*Packer et al., 2019*; *Figure 10A*). Based on the expression of some characterized reporter transgenes and their single-cell sequence analysis of the embryo, Packer et al. made tentative cell assignments for most subclusters of the L2 pharynx (see Supplemental Table 12 in *Packer et al., 2019*).

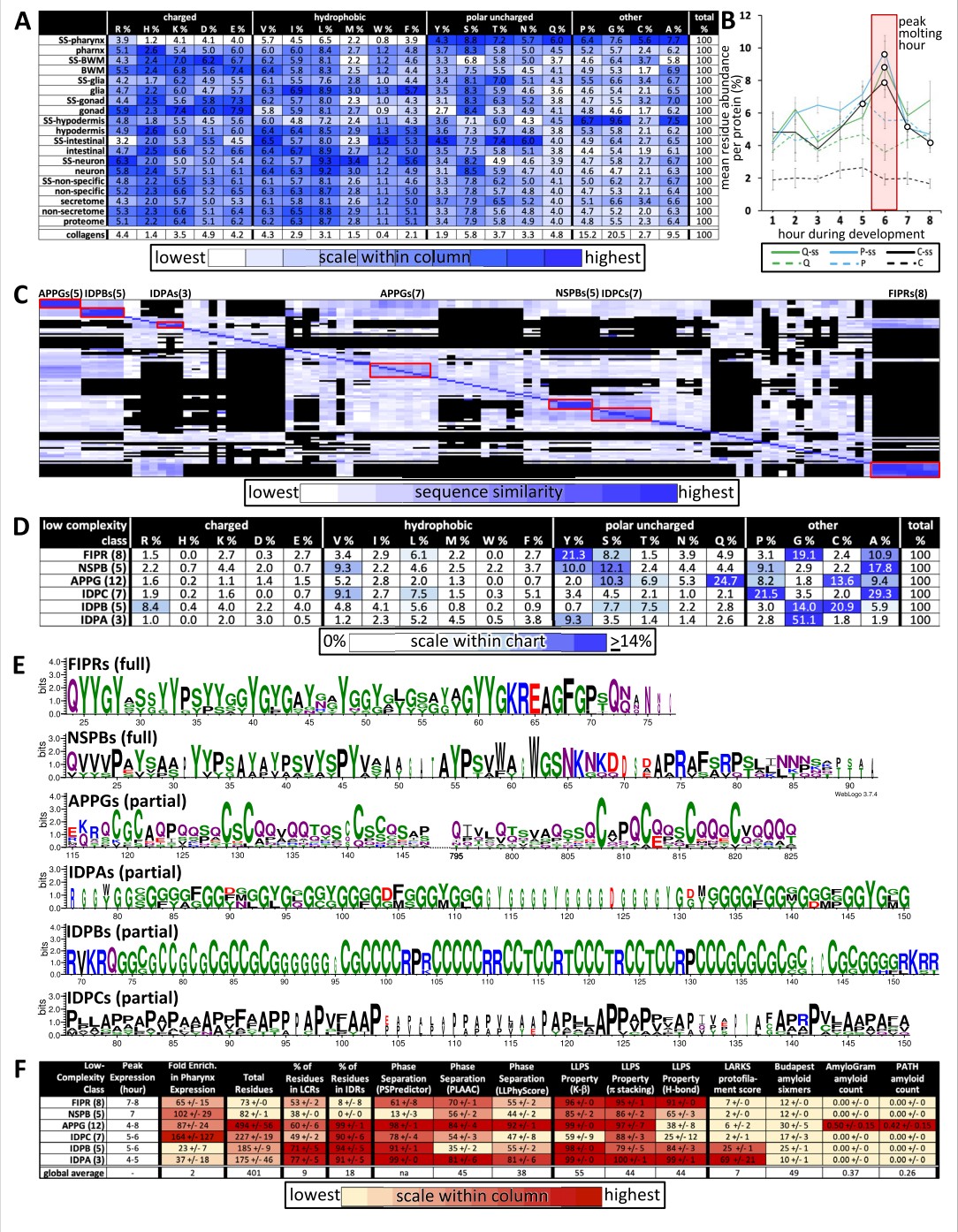

**Figure 9.** Properties of the low-complexity protein families that are likely secreted into the developing cuticle. (**A**) Average percent amino acid composition of the proteins within the indicated tissue type. The percentages along a single row sum to 100. The color scale indicates the range of values within a single column so as to compare the relative abundance of the indicated residue among the different protein sets. The collagens are not included in the color scale comparison. SS, secreted proteins based on harboring a signal sequence; BWM, body wall muscles. All of the mean residue percentages from the set of proteins secreted from the pharynx cells are significantly different compared to that of the remaining proteome (Student's *t*-test; p<2E-05). (**B**) A plot of the average percentage cysteine, proline, and glutamine composition of each protein as a function of developmental time. Secreted (ss) and non-secreted proteins are represented by solid lines and dashed lines, respectively. Open circles indicate signification differences relative to the non-secreted class (p<0.05). (**C**) Clustal Omega pairwise comparisons of all 106 low-complexity proteins in the pharynx secretome. Both X and Y axis have the same 106 proteins in the same order. Families with high sequence identity are outlined with a red box. (**D**) Similar to (**A**), except that residue composition is restricted to the indicated low-complexity family and that the color scale compares percentages across the entire chart. (**E**) Consensus sequence logos for the indicate protein families. The full consensus sequence (without the signal peptide) of the FIPRs and NSPBs is

*Figure 9 continued on next page*

Figure 9 continued

shown. The full consensus sequence of the remaining groups is given in Figure S5. (**F**) A chart of properties for the six low-complexity families. Because the PLAAC algorithm can report negative scores up to –60, 60 was added to the PLAAC scores of all gene products for the sake of clarity. All values show means ± standard error of the mean.

The online version of this article includes the following source data and figure supplement(s) for figure 9:

**Source data 1.** Supporting information for the chart diagram in *Figure 9C*.

**Figure supplement 1.** Consensus sequence for select low-complexity families.

We searched the literature for additional GFP reporter transgenes that are expressed in the postembryonic pharynx to help refine the identities of many of the L2 pharynx subclusters (*Figure 10—figure supplements 1 and 2*). We then transformed the Cao and Packer et al. L2 pharynx subcluster data into transcript summaries (see 'Materials and methods') and examined the expression level of oscillating pharynx-enriched transcripts in each of the subclusters (*Figure 10B and C*; see *Figure 1* for the relative location of each cell type).

During hours 3, 4, and 5, abundant products are secreted by the e epithelial cells, the mc3 marginal cells and presumptive pm6 and 7 transdifferentiated cells (see below). The identity of these transcripts (see *Figure 4* and *Figure 4—source data 1*) suggests that the cells are accumulating stores for the catabolism of the old cuticle and construction of the new one at the onset of the molt. Despite being confident in our assignment of cluster 11 as pm1 (*Figure 10—figure supplements 1 and 2*), the expression profile of cluster 11 is more like the arcade, e epithelial cells, and mc3 marginal cells than muscle, suggesting that pm1 may also play a role in the catabolism of the old cuticle. This is consistent with the correlation between the pharynx UMAP plot for ABU-14 and what we observe in animals with fluorescently tagged ABU-14 (*Figure 6A and A'*).

During hours 5 and 6 (which is the peak molting hour), the arcade and e epithelial cells produce abundant secreted components, consistent with the construction of a new buccal cuticle (*Figure 10B and D*). The mc1 and mc2 marginal cells also secrete abundant product (*Figure 10B and E*), again consistent with the construction of the channel cuticles and sieve (see *Figures 1, 6A and A'*).

Conspicuously absent from the expression profiles of confidently assigned subclusters is abundant secretion from the cells that surround the grinder in the posterior bulb (i.e., pm6 and pm7). Subcluster 22, which is confidently identified as pm5, pm6, pm7, and pm8 muscle, express only low levels of secreted proteins during the peak molting hour. Previous work has shown that the pm6 and pm7 cells transdifferentiate from muscle into highly secretory cells during the molting period to build a larger grinder (*Sparacio et al., 2020*). Based on the expression of a combination of markers (*Figure 10—figure supplements 1 and 2*) and the abundant expression of secreted products, we infer that subclusters 1 and 5 represent transdifferentiated pm6 and pm7 that secrete many of the same components used in the anterior pharynx epithelia to build the grinder (*Figure 10B and F*). We find that the IDPAs and IDPBs are expressed in the early transdifferentiating pm6 and pm7 cells (*Figure 10B* and *Supplementary file 2*), and therefore likely contribute to grinder formation. This prediction is consistent with our finding that disruption of IDPA-3, which localizes to the grinder (*Figure 6B and B'*), results in obvious grinder defects (*Figure 5D*). This prediction is also supported by the exclusive localization of tagged IDPB-3 to the grinder and pm6 cells (*Figure 6C and C'*). Finally, *idpb-1* and *idpp-3* are two genes belonging to subcluster 1 (*Figure 10B*, hours 4 and 5) and Yuji Kohara's mRNA in situ expression database reveals robust and specific expression of these two genes in only the posterior bulb cells (*Motohashi et al., 2006*; *Supplementary file 1*). Together, these observations are consistent with the assignment of subclusters 1 and 5 to the transdifferentiating pm6 and pm7 cells.

During the peak molting hour 6, IDPCs and the APPGs are expressed in most cells that contribute to the pharyngeal cuticle. Again, Kohara's mRNA in situ database confirms this interpretation with robust and specific pharynx expression patterns for *abu-6*, *abu-14*, *appg-2*, *idpc-1*, *idpc-3*, and *idpc-5*, and *pqn-13* (*Supplementary file 1*). The localization of tagged ABU-14 and IDPC-1 also supports this conclusion (*Figure 6A, A', D and D'*).

During hours 7 and 8, NSPB and FIPR expression is more restricted to the arcade, e epithelial cells, and the mc1 cells (*Figure 10B* and *Supplementary file 2*). Tagged NSPB-12 supports this prediction (*Figure 6E and E'*). Tagged FIPR-4, while localizing to the anterior cuticle, is also present in the posterior cuticle, suggesting that secreted FIPR-4 may be able to diffuse extensively (*Figure 6F and F'*).

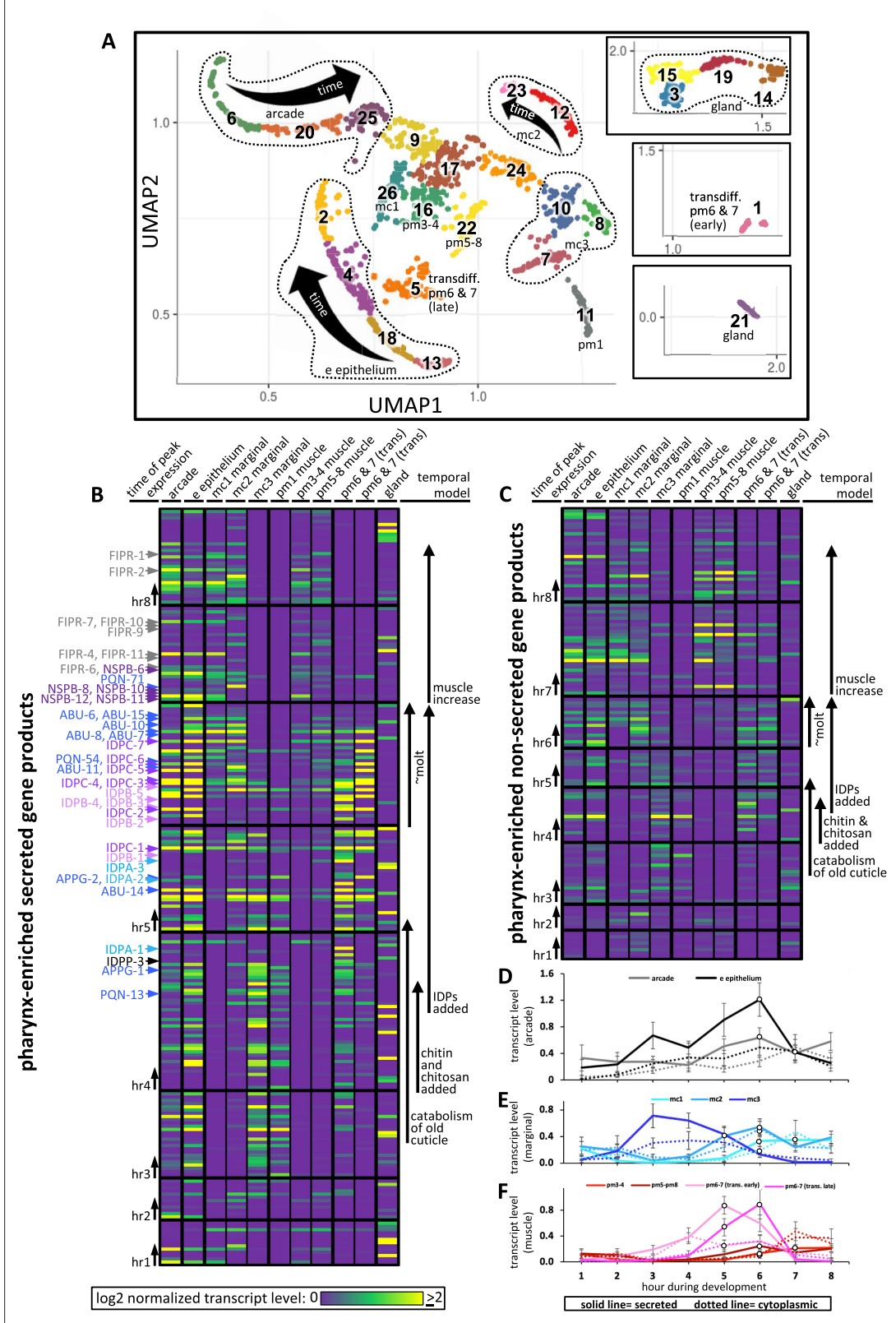

**Figure 10.** Expression of pharynx-enriched genes in distinct cell types. (**A**) A UMAP of 1675 pharynx cells modified with permission from *Packer et al., 2019*'s online tool. The clusters are numbered according to *Packer et al., 2019*. The cell type identities are partially based on those from *Packer et al., 2019* (see *Figure 10—figure supplements 1 and 2* for details). Due to space constraints, three cluster groups from the map are shown as insets. (**B, C**) The expression level of the pharynx-enriched gene set in the indicated tissue type. The graph notation and the order of genes in rows is preserved

*Figure 10 continued on next page*

*Figure 10 continued*

from *Figure 4A*. Genes encoding a signal peptide are shown in (**B**) and those without a signal sequence are shown in (**C**). The mc1, pm3-4, and pm5-8 values represent the average gene expression of the cells within the respective clusters (26, 16, and 22). The values corresponding to the other cell types represent the highest average from among the group of clusters that constitute that cell type. For example, the arcade cells are represented by clusters 6, 20, and 25, but the expression level from each of these clusters is distinguished by time, not space, and averaging signal from all three would dilute the expression level that represents that cell type. All members of the six low-complexity families are indicated on the left of (**B**) and the color code is the same as that present in *Figure 4A*. (**D–F**) The average transcript level of all genes within the indicated cell type as a function of binned time. Open white circles represent a significantly greater value (p<0.01) compared to the bin 2 hr previous.

The online version of this article includes the following figure supplement(s) for figure 10:

**Figure supplement 1.** Identity assignment of the pharynx UMAP clusters.

**Figure supplement 2.** UMAP plots of the gold standard genes used to assign identity to the pharynx UMAP reference cluster.

Cytoplasmic components involved in muscle development peak in expression during hours 7 and 8 (*Figure 10C and F*).

The number of genes expressed from the gland cells is not obviously enriched in any one temporal interval (*Figure 10B and C*), yet the overall abundance of gland transcripts peak in hour 5 (*Figure 4D*). This apparent contradiction is due to the two most abundantly expressed genes from the gland, *phat-2* and *phat-4*, peaking in expression during hour 5 (*Figure 10B*, *Figure 4—source data 1*). PHAT-2 and PHAT-4 are paralogous mucin-like proteins (*Ghai et al., 2012*; *Smit et al., 2008*) whose timing of peak expression suggests that they may play a role in cuticle structure or function. PHAT-2 and PHAT-4 notwithstanding, the overall temporal pattern of expression from the gland suggests that its products do not play a large role in cuticle turnover during the molt.

## Discussion
### A model of pharyngeal cuticle construction

Here, we have mined published resources to bioinformatically reconstruct the *C. elegans* pharynx cuticle. This map provides unprecedented insight into the spatiotemporal progression of cuticle construction. During hours 3 and 4, genes that encode homologs of chitin and amyloid catabolic enzymes peak in their expression. These include the predicted chitinases CHT-1, CHT-2, CHT-5, CHT-6, two predicted amyloid peptidases (NEP-1 and NEP-12) (*Iwata et al., 2001*), and the NAS-6 protease that helps degrade pharyngeal cuticle (*Sparacio et al., 2020*; *Park et al., 2010*). The predicted amyloid-fibril inhibitor ITM-2 (*Cohen et al., 2015*) also peaks in expression during this interval, perhaps to prevent aggregation during disassembly. The expression profile at this interval is consistent with preparation for apolysis (the detachment of the old cuticle).

During hours 4, 5, and 6, anabolic enzymes and constructive components peak in expression. These include the characterized chitin synthase CHS-2 (*Zhang et al., 2005*), putative chitosan synthases LGX-1 and CHTS-1 that deacetylates chitin to produce chitosan (*Heustis et al., 2012*), and putative chitin binders and cross-linkers CHTB-1, CHTB-2, and CHTB-3. In this interval, components implicated in amyloid metabolism also peak in expression. These include a predicted amyloid chaperone LRX-1 (*Cam et al., 2004*), two predicted amyloid-chitin linkers LRPC-1 and PQN-74 (*Brodeur et al., 2012*), and a predicted amyloid precursor protein interactor FEH-1 (*McLoughlin and Miller, 2008*).

During hours 5 and 6, a massive increase in gene expression of the pharynx secretome occurs. The period coincides with the upregulation of secreted intrinsically disordered proteins from the pharynx epithelium and includes successive waves of peak transcript expression encoding four of the intrinsically disordered families, IDPA, IDPB, IDPC, and APPG members that have been previously implicated in cuticle development (*George-Raizen et al., 2014*).

During hours 5 and 6, the gene products that peak in expression are rich in PPIs compared to the proteins secreted by other tissues. The protein interactors within the pharynx secretome network are highly enriched in low-complexity sequences predicted to phase separate.

Finally, during hours 7 and 8, genes that encode muscle contraction components are upregulated, which likely corresponds to a period of tissue growth at the tail end of molting. We also see the peak expression of the low-complexity families NSPB and FIPR, which are likely added to the cuticle in its final phase of maturation. Together, these observations illustrate the utility of the spatiotemporal map in revealing the logic by which a cuticle is assembled.

## The pharynx cuticle is unlikely to harbor amyloid fibrils

Despite the pharynx secretome not being *enriched* for amyloidogenic proteins, multiple pharynx cuticle proteins are predicted to nevertheless be amyloidogenic. In addition, multiple predicted amyloid regulators are upregulated during pharyngeal cuticle development. Yet, evidence argues against the presence of amyloid fibrils within the pharynx cuticle. We speculate that fibril formation may not occur within the pharyngeal cuticle because of the heterogeneous mixture of the IDR-rich proteins within the structure. In other words, the relatively low concentration of any one protein species within the cuticle mixture may preclude the assembly of long fibrils with birefringent properties. Indeed, the presence of other IDRs antagonizes Abeta42 fibril formation (*Ikeda et al., 2020*). A second factor that may antagonize fibril formation is the presence of a chitin matrix. During the formation of the squid beak, IDR-rich proteins form phase-separated coacervates that infiltrate a chitin matrix (*Tan et al., 2015*), which may limit amyloid fibril formation. It is unknown whether similar dynamics take place during pharyngeal cuticle development. Third, the pharynx secretome is enriched with kinked β-structure that can support liquid-phase separation and may facilitate protofilament formation but otherwise antagonizes extensive fibril growth (*Hughes et al., 2018*). Notably, many well-characterized proteins with amyloidogenic propensity only form fibrils when associated with pathogenesis (*Patel et al., 2015*; *Cremades et al., 2012*).

The idea that the pharyngeal cuticle contains a non-rigid network of IDRs is appealing because the pharyngeal cuticle must be sufficiently flexible to accommodate pharynx movements along the dorsal–ventral (*Huang et al., 2008*) and anterior–posterior (*Avery, 1993*) axes. Indeed, others have suggested that IDR-rich proteins within chitin-based cuticles might add elastic properties to what might otherwise be an inflexible chitin-based material (*Andersen, 2011*). An elastic cuticle might also aid in returning the open and extended lumen (which results from pharynx muscle contraction) to the relaxed ground state position.

## Potential contributions of IDPs to the cycles of cuticle formation and destruction

A key feature of phase-separating IDRs is their potential to reversibly transition between different states of matter depending on local conditions and post-translational modifications (*Murray et al., 2017*; *Deiana et al., 2019*), including liquids and gel-like biomaterials. The pharyngeal cuticle must soften, be shed, and be reconstructed about every 8 hr during larval development (*Lazetic and Fay, 2017*). The notion that a network of IDR-rich proteins is not locked into a rigid state but may instead be regulated to increase or decrease intermolecular interactions and change material properties as needed during the molting cycle is an appealing idea that requires further investigation.

Both the APPGs and the IDPBs are highly enriched with cysteines and contribute heavily to an increase in the relative abundance of cysteines that is likely deposited into the developing cuticle as the animal prepares to molt. Other work has shown that the *C. elegans* cuticle is indeed rich in disulfides during the intermolt period and becomes reduced to facilitate apolysis (*Stenvall et al., 2011*). Furthermore, exogenously supplied reducing agent can induce pharyngeal cuticle apolysis during the intermolt period (*Stenvall et al., 2011*). Manipulating the redox state of cysteines can alter the ability of IDR-rich proteins to phase separate or further condense (*Reed and Hammer, 2018*; *Zhang et al., 2020*; *Kato et al., 2019*). Whether the abundant cysteines within the pharyngeal cuticle are key to phase separation and yield a network of variably dynamic cross-linked proteins remains to be determined.

The spatiotemporal map suggests that many different types of IDPs likely contribute to the pharyngeal cuticle. Previous studies have shown that coexisting condensed protein phases, each with distinct protein compositions, can yield complex biomaterials with layers and other non-uniform properties (*Mountain and Keating, 2020*; *Lu and Spruijt, 2020*; *Lin et al., 2018*). The distinct compositions of the six families uncovered by the spatiotemporal map are suggestive of the potential immiscibility of their condensed phases and of physical mechanisms for building the cuticle, particularly when combined with varying temporal expression, similar to what is observed during cuticle formation of the mussel byssus (*Jehle et al., 2020*). What is becoming clearer is how evolution has repeatedly capitalized on biomolecular condensates to make complex protective structures.

## The molecular composition of cuticles may be evolutionarily plastic

The extent to which the blueprint of *C. elegans* pharyngeal cuticle development is conserved among other phyla within Ecdysozoa is unknown. The incorporation of chitin and chitosan within Ecdysozoan cuticles is firmly established (*Moussian, 2010*; *Muthukrishnan et al., 2019*). Mounting evidence also indicates that the arthropod cuticle has abundant IDR-rich proteins (*Andersen, 2011*) with amyloid-like folds (*Sviben et al., 2020*). However, of the 12 families of known arthropod cuticle proteins, only CPAP1 and CPAP3 have recognizable conservation with nematodes (*Willis, 2010*; *Muthukrishnan et al., 2019*). CPAP1/3 are defined by the ChtBD2 chitin-binding domain that is also harbored in the pharyngeal cuticle proteins CHTB-2, LRPC-1, and PQN-74. CPR is the only other arthropod cuticle family protein beyond the CPAPs that is well-characterized to bind chitin; the function of the remaining families remains obscure (*Willis, 2010*; *Muthukrishnan et al., 2019*). Furthermore, homologs of the six low-complexity families found within the pharyngeal cuticle cannot be found beyond Nematoda. It is not clear whether the IDR-rich proteins of arthropod and nematode cuticles are of distinct evolutionary origin or have simply diverged beyond recognition because of reduced primary sequence constraints. Regardless, the IDP-chitin combination clearly provides an effective barrier that is evolutionarily malleable to provide diverse form for millions of species.

## The spatiotemporal map is a foundation for future investigation

The spatiotemporal map provides a starting point to investigate many important questions. First, what is the mechanism by which the temporal unfurling of gene expression is coordinated? While the global oscillatory pattern of *C. elegans* gene expression has been modeled in detail (*Meeuse et al., 2020*; *Hutchison et al., 2020*), how the oscillatory pattern of each gene becomes temporally offset from other oscillating genes is not understood. One candidate regulator of oscillation is the *C. elegans* period ortholog LIN-42. LIN-42 is a known regulator of developmental timing in the worm (*Jeon et al., 1999*; *McCulloch and Rougvie, 2014*), is expressed in the pharynx and other tissues (*Monsalve et al., 2011*), and alters the timing of molting when disrupted (*Monsalve et al., 2011*). Temporally uncoordinated gene expression would almost certainly be lethal, yet *lin-42* null mutants are viable (*Edelman et al., 2016*), suggesting that other key regulators are involved. Investigating the relationship between tissue-restricted transcription factors and their targets as a function of developmental time may provide insight into the coordinated temporal regulation of gene expression (*Roy, 2022*).

Second, how are catabolic and anabolic processes separated and regulated? The process of molting leaves animals vulnerable and must occur rapidly. In that light, it is perhaps not surprising that we observe a temporal overlap of expression of catabolic and anabolic components. Previous work on the ultrastructure of the grinder cuticle and molt indicates that dense core vesicles (DCVs) lie in wait until the new cuticle is assembled, at which point the DCVs likely fuse with the plasma membrane and dump their contents (*Sparacio et al., 2020*). Based on the timing of the peak expression of secreted components with respect to the timing of the molt itself, we surmise that (1) there is a temporal lag between the period of peak expression for a given gene and when protein abundance peaks, and (2) unknown mechanisms regulate the timing at which catabolic and anabolic components, perhaps within distinct DCVs, are released into the ECM. In this way, it might be possible to have temporal overlap in the peak expression of genes that encode catabolic and anabolic components. Exactly how the secretion of catabolic and anabolic components is regulated remains to be determined.

Finally, how are patterns within the pharyngeal cuticle established? Cuticle lumen shape and size are likely patterned by the underlying cells, but this simply extends the question. How is the patterning of the electron-dense cuticle ribbing established? Is the information that governs pattern of the flaps, which is seemingly independent of the shape of nearby cells, contained within the flaps' protein components? Do the successive waves of expression of low-complexity protein families contribute to the layering of the cuticle seen in the electron micrograph cross sections? How might coexisting condensed phases of these proteins establish layering and other complexities of the cuticle structure? The spatiotemporal map of pharyngeal cuticle construction presented here may serve as the foundation for answering these and other questions in the future.

# Materials and methods

## Methods

### *C. elegans* culture, microscopy, and synchronization

*C. elegans* strains were cultured as previously described (*Kamal et al., 2019*). Unless otherwise noted, the wildtype N2 Bristol strain was used. Worms are prepared for imaging by washing them three times in M9 buffer and resuspended in a paralytic solution of either 50 mM levamisole or 50 mM sodium azide. The resuspended worms are then mounted on a 3% agarose pad on a glass slide and a coverslip for all brightfield and fluorescent microscopic analyses and photography. Unless otherwise noted, a Leica DMRA compound microscope with a Qimaging Retiga 1300 monochrome camera was used for routine analyses. Confocal imaging was performed using the Zeiss LSM 880 attached to an inverted epifluorescent microscope with a ×63 (numerical aperture 1.4) oil immersion objective. Worms expressing GFP were excited using an argon laser operating at 488 nm. Confocal images were obtained using digital detectors with an observation window of 490–607 nm (green). Pseudo-transmission images were obtained by illuminating with the 488 nm laser and detected with the transmission photomultiplier tube and converted to digital images. Birefringent analyses were done with the Leica DMRA with the polarizer and analyzer polarized filters at right angles to one another. Colored birefringence images were captured using a Leica Flexacam C1 colour camera.

Synchronized populations of worms were obtained by first washing off a population of worms rich with gravid adults on plates with M9 buffer, collecting the sample in 15 mL conical tubes, and centrifuging the samples at 800 × *g* to concentrate worms. The supernatant is then removed via aspiration and additional washes with M9 buffer are done until all bacteria are removed. 1.5 mL of suspended worms are then left in each tube and in rapid succession, 1 mL of 10% hypochlorite solution (Sigma) is added followed by 2.5 mL of 1 M sodium hydroxide solution and 1 mL double-distilled water. The mixture is incubated on a nutator for ~3.5 min. The tubes are then vortexed for 10 s with two 5 s pulses and visually inspected for near-complete digestion of post-embryonic worms. M9 buffer is then added to 12 mL. The tube is spun at 2000 rpm for 1 min, supernatant removed, fresh M9 buffer added to ~12 mL, and the tube is vigorously shaken. This is repeated two more times. After the final wash, the tube is incubated overnight on a nutator at 20°C to allow egg-hatching. The next day, the sample is checked for synchronized L1s. To obtain other synchronized stages, the synchronize L1s are plated on solid agar substrate with *Escherichia coli* food and allowed to progress to the desired stage before processing.

### *C. elegans* transgenes

NQ824 *qnEx443[Pabu-14:abu-14:sfGFP; rol-6(d); unc-119(+)]* was a kind gift from David Raizen. We chromosomally integrated the *qnEx443* extra-chromosomal array using previously described methodology (*Mello and Fire, 1995*), resulting in the RP3439 *trIs113[Pabu-14:abu-14:sfGFP; rol-6(d); unc-119(+)]* strain. Tagged IDPC-1 was generated by InVivoBiosystems (Eugene, USA) by using CRISPR/ Cas9-based mGreenLantern knock-in at the C-terminus of the Y47D3B.6 native locus. Two guide RNAs, sgRNA1 (5′-AGCTCCTGGGACACAGGCTG-3′) and sgRNA2 (5′-GCTGGAGTCTGCCAGTGCGC-3′), were designed to target the C-terminus of Y47D3B.6. The single-stranded donor homology DNA included 35 bp homology arms flanking a GGGSGGGGS linker and the mGreenLantern sequence. Insertion of the mGreenLantern sequence was identified by PCR and confirmed by sequencing.

IDPA-3, IDPB-3, FIPR-4, and NSPB-12 were tagged C-terminally with mNeonGreen. The mNeonGreen coding sequence was PCR-amplified from the *C. elegans* strain WD835 (a kind gift from Brent Derry) using the following primers: 5-mNeon (5′-GTCAGACCGGTGGCGGTGGATCAGTCTC CAAGGGAGAGGAGGACAACATGG-3′) and 3-mNeon (5′-TTACGGAATTCTCACCCTTGTAGAGCTC GTCCATTCCCATG-3′). The 5-mNeon primer introduced a flexible GGGGS linker sequence to the epitope tag. The resulting PCR product was purified, digested with AgeI and EcoRI, and the 728 bp fragment was ligated to the 5 kb AgeI/EcoRI digested pPRGS762 (*unc-6p*::YFP) vector backbone to generate pPRJK1199 (*unc-6p*-mNeonGreen-*unc-54* 3′UTR). The coding and upstream promotor sequences (up to the end of the upstream gene) of IDPA-3, IDPB-3, FIPR-4, and NSPB-12 were amplified from wildtype *C. elegans* N2 genomic DNA template using the following primer pairs: 5-IDPA-3 (5′-CCGTACTGCAGAGCATCTCTAGAACTGACCATCTGACC-3′) and 3-IDPA-3 (5′-GTTAGACCGGTG TTTGGCATTGGTGGCCATCCTCCTTG-3′); 5-IDPB-3 (5′-CAGTACTGCAGAGCAGATGATCTCACTA

GTGCAACC-3') and 3-IDPB-3 (5'-GTTAGACCGGTGCACTTGTCTCCTCCCTTGGCTGG-3'); 5-FIPR-4 (5'-CCGTACTGCAGCATGTGTTGGTTTTGTCATAGAAACTGTCG-3') and 3-FIPR-4 (5'-GTTAGACC GGTGTTCTGAATAGGTCCAAATCCAGC-3'); 5-NSPB-12 (5'-CCGTAATGCATTTGCTGGCGTATTGTCT AAACCTTGC-3') and 3-NSPB-12 (5'-GTTAGACCGGTAGCGGTGGTTGGCTTCTGATTGTTAAG-3'). The PCR products were purified, digested with PstI and AgeI (IDPA-3, IDPB-3, FIPR-4) or NsiI and AgeI (NSPB-12), and ligated to the 4.2 kb fragment of the PstI/AgeI digested pPRJK1199 vector to generate pPRJK1213 (idpa-3p::IDPA-3::mNeonGreen [1232 bp of sequence upstream of the ATG]), pPRJK1202 (idpb-3p::IDPB-3::mNeonGreen [334 bp of sequence upstream of the ATG]), pPRJK1212 (fipr-4p::FIPR-4::mNeonGreen [1360 bp of sequence upstream of the ATG]), and pPRJK1203 (nspb-12p::NSPB-12::mNeonGreen [1973 bp of sequence upstream of the ATG]), respectively. All constructs were verified by sequencing. Wildtype *C. elegans* N2 worms were injected with each of the constructs described above along with the pPRGS382 (*myo-2p*::mCherry) co-injection marker at the following concentrations for expression analysis: pPRJK1213 (10 ng/µL) + pPRGS382 (2 ng/µL) + pKS (88 ng/µL); pPRJK1202 (10 ng/µL) + pPRGS382 (2 ng/µL) + pKS (88 ng/µL); pPRJK1212 (10 ng/µL) + pPRGS382 (2 ng/µL) + pKS (88 ng/µL); pPRJK1203 (10 ng/µL) + pPRGS382 (2 ng/µL) + pKS (88 ng/µL).

## Pulse-chase analyses

Synchronized wildtype L1 worms are plated on 10 cm plates at 7000 L1s/plate seeded with OP50 *E. coli* strain. Plates with worms destined for pulse-chase analyses of larvae or adults are grown at 16°C or 25°C, respectively. Then, 72 hr after plating, the 'L3' samples and the 'adult' samples are washed with M9 to remove bacteria. The concentrations and solvents for all dyes are described in the relevant methods section. In all cases, 50 µL of packed worms from centrifugation are used per tube in the dye incubation. Note that the number of worms should not exceed 1000 because adding more worms reduces stain intensity. Also, siliconized tips are used with the ends cut with flame-sterilized scissors to avoid injuring the worms. The tubes with worms and dye are then incubated on a nutator for 3 hr in the dark at room temperature. After incubation, the 1.5 mL tubes are spun at 5000 rpm for 1 min and the concentrated pellet is carefully transferred to 15 mL falcon tube and washed with 8 mL of M9 buffer to remove excess dye. The tubes are inverted gently and spun at 2000 rpm for 1 min. The supernatant is removed and the concentrated washed worms are spotted onto the clear (agar) surface of 6 cm plates seeded with OP50. Then, 30 min later, 20–30 worms are picked onto a second plate lightly seeded with OP50. The staining of the cuticle for each is then semi-quantitatively assessed on an epifluorescent microscope. These data represent the pre-chase counts. The scoring system was as follows: animals exhibiting robust staining in the buccal cavity and anterior channels = 3; animals exhibiting moderate staining in the buccal cavity and anterior channels = 2; animals showing faint staining in the buccal cavity and anterior channels = 1; animals showing no detectable staining in any part of the pharynx cuticle = 0. The remaining animals on the original 6 cm plate are incubated for a total of 18 hr at 20°C, after which dye staining of the cuticle is quantified. These data represent the post-chase counts.

### Generating *mlt-9(RNAi)* Cuticle Defects

*mlt-9* RNAi was carried out as described previously (**Frand et al., 2005**) with some modifications. Briefly, a bacterial culture expressing dsRNA of *mlt-9* (referred to here as *mlt-9(RNAi)*) (**Kamath et al., 2003**) was started from a single colony in 30 mL LB broth containing 100 µg/mL ampicillin for 18 hr at 37°C at 200 rpm. The cells were pelleted by centrifuging at 3200 rpm for 15 min, after which the cells were concentrated tenfold. Then, 1 mL of the pelleted cells was added to 10 cm NGM agar plates containing 8 mM IPTG and 40 µg/mL carbenicillin and left to dry overnight at room temperature in the dark. The next day (day 0), 6500 synchronized L1s were plated onto each RNAi plate, after which the plates were stored at 16°C in the dark. Ninety hours later, the worms were inspected for *mlt-9* RNAi phenotypes. Approximately 50% of *mlt-9(RNAi)*-treated worms exhibit the expected cuticle defects. Performing mock RNAi with the empty L4440 plasmid failed to yield worms with obvious cuticle defects.

## Dye staining of wildtype and *mlt-9*(RNAi) animals

### Congo Red (CR) staining

Synchronized wildtype adult worms were washed and incubated with 0.02% CR from a 1% stock (w/v, dissolved in DMSO; Fisher chemical C580-25; CAS 573-58-0) in 500 µL of liquid NGM for 3 hr in the dark. Worms are then prepped for microscopic analysis as described above.

### Thioflavin S (ThS) staining

Synchronized wildtype adult worms were washed and incubated with 0.1% ThS from a 10% stock (w/v, dissolved in DMSO; ThS; SIGMA, T1892-25G) in 500 µL of liquid NGM for 3 hr in the dark. Worms are then prepped for microscopic analysis as described above. The concentration chosen for ThS staining of *C. elegans* pharynx was based on a published protocol (*Wu et al., 2006*). ThS is a complex mixture of molecules with two major species of 377.1 and 510.1 MW and several other minor species (*Enthammer et al., 2013*). Given that the ratio of molecules is unknown, we used an average MW of 443.6 for ThS in our calculations.

### Eosin Y (EY) staining

EY staining was performed as described (*Heustis et al., 2012*). Briefly, synchronized wildtype adult worms were washed and incubated with 0.15 mg/mL from a 5 mg/mL stock (dissolved in 70% ethanol; Eosin Y; Sigma-Aldrich, E4009) in 500 µL of liquid NGM for 3 hr in the dark. Worms are then prepped for microscopic analysis as described above. Note that eosin Y stock should be stored at –20°C and before its use it should be incubated at 55°C for ~2 min and vigorously vortexed to ensure its solvation.

### Calcofluor white (CFW) staining

Synchronized wildtype adult worms were washed and incubated with 0.005% CFW from a 1% stock (w/v, dissolved in DMSO; Fluorescent Brightener 28, Sigma-Aldrich, CAS 4404-43-7) in 500 µL NGM for 3 hr in the dark. Worms are then prepped for microscopic analysis as described above. Note that the CFW stock should be placed in boiling water for ~2 min and then vigorously vortexed to ensure solvation of the dye.

## Calculations of low-complexity and intrinsic disorder

LCRs in the amino acid sequences of each protein within the *C. elegans* proteome (WormBase release WS274) were identified using the SEG algorithm with default stringency parameters set (i.e., WINdow = 12, LOWcut = 2.2, HIGhcut = 2.5) (*Wootton and Federhen, 1993*). Percentage sequence in LCRs was calculated for each protein based on the total number of residues found within LCRs returned by SEG relative to protein length. The intrinsic disorder of each protein within the *C. elegans* proteome (obtained from WormBase version WS274) was analyzed using the Spot-Disorder script (*Hanson et al., 2017*). The computational analysis was conducted using the Niagara supercomputer at the SciNet HPC Consortium. The GNU 'parallel' package was used to perform the computational analysis in parallel. The individual protein SPOT-Disorder output data were then computationally analyzed using Python for IDRs (defined as any string of 30 or more disordered residues), total number of disordered residues, and percentage of amino acid residues within intrinsically disordered regions.

## LLPhyScore calculations

The LLPhyScore phase separation score of each protein was calculated using the LLPhyScore algorithm (*Cai et al., 2022*). The LLPhyScore algorithm is a machine learning-based interpretable predictive algorithm that is based on the idea that a combination of multiple different physical interactions drives protein liquid–liquid phase separation. A protein's LLPhyScore is a weighted combination of eight sub-scores, each representing one physical feature that is inferred from the input sequence. These physical features include protein–water interactions, hydrogen bonds, pi–pi interactions, disorder, kinked-beta structure, and electrostatics. The scores are optimized via training with 500+ experimentally known phase-separating protein sequences against selected negative sequences. More details about this algorithm can be found in the manuscript in preparation.

## AmyloGram and path analyses

AmyloGram (*Burdukiewicz et al., 2017*) is a method based on machine learning, trained on hexa-peptides experimentally tested for their amyloidogenic propensities (*Wozniak and Kotulska, 2015*). Amino acids are represented by the alphabet that best encoded amyloidogenicity of peptides modeled by n-grams, and it was optimized by a random forest classifier. Classification of a protein amyloidogenicity included calculating its profile with a hexapeptide window shifting along the protein chain. Proteins with amyloid propensity were identified on the basis of an appearance of at least one amyloidogenic fragment. To avoid an excessive number of false positives, non-default specificity values were used: 0.95 and 0.99.

PATH (*Wojciechowski and Kotulska, 2020*) uses molecular modeling and machine learning. It is a computational pipeline based on Python and bash scripts, using Modeller (*Sali and Blundell, 1993*) and PyRosetta (*Chaudhury et al., 2010*). A potentially amyloidogenic query sequence of a hexa-peptide was threaded on seven representative amyloid templates. Comparative structure modeling provided evaluation of the models with statistics and physics-based functions. Next, the scores were used by the logistic regression classifier. The analyses with PATH were carried out in two stages. The first scan along the protein chain was done by AmyloGram with the specificity threshold at 0.99, which was then followed by structural modeling and classification using PATH. The second stage was only applied to amyloid-positive regions found by AmyloGram.

## LARKS analyses

LARKS predictions were done on a proteome downloaded from WormBase on October 18, 2021. Sequences not completely comprised of the 20 canonical amino acids were rejected from analysis. Each protein from the filtered proteome set of 20,042 proteins was then submitted for LARKS predictions. First, the sequence was separated into a series of overlapping hexapeptide segments (each segment overlapped with five residues from the segment before it; a 150 amino acid sequence contains 145 hexapeptides). The sidechains for each residue in a hexapeptide are computationally grafted onto a fibril model for each of three different LARKS structures (FUS-SYSGYS, FUS-STGGYG, and hnRNPA1-GYNGFG; PDB IDs: 6BWZ, 6BZP, and 6BXX). Energy minimization is done using a Rosetta energy score as a readout, and if the final energy is below a backbone-dependent threshold, then hexapeptide segment is considered a LARKS. Proteins' LARKS content was determined by the number of favorable LARKS segments divided by the length of the protein.

## In vitro expression and analysis of IDPs

### Expression vectors and constructs

All protein expression vectors generated for this work were derivatives of the pMBP-FUS-FL-WT (a gift from Nicolas Fawzi [Addgene plasmid # 98651; http://n2t.net/addgene:98651; RRID:Addgene_98651; *Burke et al., 2015*], which was modified to remove the FUS1 coding region and to have two cloning sites BamHI and NotI) for facile cloning of new proteins in phase with the HIS-tagged Maltose Binding Protein (MBP) at the N-terminus followed by a TEV protease cleavage site (TEVcs) to generate pPRRH1197. The coding region of proteins of interest (minus signal sequences) was codon optimized for expression in *E. coli*, synthesized with appropriate linkers, and subcloned into frame with MBP (GenScript), resulting in pPRPM1191 (HIS::MBP::TEVcs::IDPC-2).

### Protein preparation and purification

Proteins were expressed in *E. coli* BL21DE3 RIPL in LB with kanamycin and chloramphenicol. Cells were grown to $OD_{600}$ of 0.5, induced with 0.5 mM IPTG, and grown overnight at 18°C. The next day cultures were centrifuged at 5000 × *g* at 4°C for 10 min. Pellets were frozen at –80°C then thawed and resuspended in lysis buffer (2.5 mM Tris pH 7.5, 500 mM NaCl, 20 mM imidazole, 2 mM DTT and 1x Protease inhibitor cocktail; Sigma, P8849). This suspension was sonicated to lyse *E. coli* and clarified by centrifugation at 39,000 × *g* for 45 min at 4°C. The cleared supernatant was added directly to a pre-equilibrated nickel column. Optimal wash and elution conditions had to be determined empirically for each protein. Purified fractions where then dialyzed with 2.5 mM Tris pH 7.5, 150 mM NaCl, 2 mM DTT to remove excess salts and imidazole and protein concentration determined with Bradford assay.

## Phase separation assays

Proteins were incubated in 2.5 mM Tris pH 7.5, 150 mM NaCl, 2 mM DTT with either 5% Ficoll (Sigma, F2637) for MBP::FUS1 or 15% Ficoll for MBP::IDPC-2 for 1 hr at 30°C with or without TEV protease (10 units in a 50 µL reaction). The optimal percent Ficoll was determined empirically. Turbidity was measure at 395 nm with a Clariostar plate reader (Mandel). 10 µL of each reaction was spotted onto slides with coverslips then condensates visualized with DIC using a Leica DMRA2 microscope at ×63 magnification.

## Protein sequence analysis and logo generation

We used Clustal Omega (*Sievers et al., 2011*) to align the 110 low-complexity protein sequences and generate a percent identity matrix based on the multiple sequence alignment. For those low-complexity proteins with a predicted signal peptide, the first 20 amino acids were removed from the protein sequence before alignment.

To generate sequence logos, full-length protein sequences from each of the low-complexity protein families identified by the percent identity matrix were aligned using ClustalW (*Thompson et al., 1994*). Sequence logos were constructed based on these alignments using WebLogo 3.7.4; (https://weblogo.berkeley.edu/; *Crooks et al., 2004*; *Schneider and Stephens, 1990*). Amino acid residues were colored according to their chemical properties: polar (G,S,T,Y,C) in green, neutral (Q,N) in purple, basic (K,R,H) in blue, acidic (D,E) in red, and hydrophobic (A,V,L,I,P,W,F,M) in black. The height of the symbol within each stack indicates the relative frequency of that amino acid in that position. Stack widths are scaled by the fraction of symbols in that position (positions with many gaps are narrow). Details of protein sequences used can be found in *Figure 4—source data 1*.

## Statistics and graphs

Except where indicated, statistical differences were measured using a two-tailed Student's *t*-test. Plots were either generated using Prism 8 graphing software or Excel.

## Materials availability statement

The *C. elegans* strains expressing the fluorescently tagged fusion proteins will be made available at the *C. elegans* Genetic Center.

## Acknowledgements

We are grateful to David Raizen and Fred Keeley for helpful conversations, and for the work of JG White, E Southgate, JN Thomson, and S Brenner, who generated the serial sections that we show in Figure 1, and KA Wright, JN Thomson, who generated the transverse images that we show in Figure 3. We thank John White and Jonathan Hodgkin for allowing MRC/LMB archival TEM images to be sent to WormAtlas (David Hall) at Albert Einstein College of Medicine for long-term curation. We thank David Hall and Zeynep Altun for helpful advice and for sharing unpublished images via WormImage and WormAtlas (funded by an NIH grant [OD 010943] to DH Hall). We also thank Iva Pritisanac for mentorship of intrinsic disorder calculations, the staff at Wormbase for assembling and communicating proteome files to us, and Tim Schedl for guidance on new gene assignments. For mutant strains, we are grateful to Mei Zhen and Wesley Hung, Harald Hutter, the *C. elegans* Gene Knockout Consortium, Don Moerman's and Bob Waterston's million mutation project, Shohei Mitani, and the *C. elegans* Genetics Centre. We thank Brent Derry and Matthew Eroglu for the codon-optimized mNeonGreen sequence. PJR dedicates this article to the recently retired Don Moerman – many things would have been more difficult without you. Funding was from NKFI grant 127909 (KT, VG), National Science Foundation Grant 1616265 (MPH), a grant (2019/35/B/NZ2/03997) from the National Science Centre, Poland (MK) NSERC Alexander Graham Bell Canada Graduate Scholarship (JK) Canadian Institutes of Health Research grants 376634 and 313296 (PJR) Canadian Research Chair grant (PJR).

## Additional information

### Funding

| Funder | Grant reference number | Author |
|---|---|---|
| NKFI | 127909 | Kristóf Takács |
| National Science Foundation | 1616265 | Michael P Hughes |
| National Science Centre, Poland | 2019/35/B/NZ2/03997 | Malgorzata Kotulska |
| Canadian Institutes of Health Research | 376634 | Peter J Roy |
| Canadian Institutes of Health Research | 313296 | Peter J Roy |
| National Science and Engineering Council of Canada | | Jessica Knox |
| Canada Research Chairs | | Peter J Roy |

The funders had no role in study design, data collection and interpretation, or the decision to submit the work for publication.

### Author contributions

Muntasir Kamal, Investigation, Visualization, Methodology, Writing - original draft, Writing – review and editing; Levon Tokmakjian, Investigation, Visualization, Methodology, Writing – review and editing; Jessica Knox, Data curation, Investigation, Visualization, Methodology, Writing – review and editing; Peter Mastrangelo, Investigation, Methodology, Writing – review and editing; Jingxiu Ji, Investigation, Visualization; Hao Cai, Jakub W Wojciechowski, Kristóf Takács, Xiaoquan Chu, Formal analysis; Michael P Hughes, Formal analysis, Methodology, Writing – review and editing; Jianfeng Pei, Vince Grolmusz, Malgorzata Kotulska, Formal analysis, Supervision; Julie Deborah Forman-Kay, Conceptualization, Supervision, Methodology, Writing – review and editing; Peter J Roy, Conceptualization, Data curation, Formal analysis, Supervision, Funding acquisition, Investigation, Visualization, Methodology, Writing - original draft, Project administration, Writing – review and editing

### Author ORCIDs

Jessica Knox ⓘ http://orcid.org/0000-0003-1465-5852
Jingxiu Ji ⓘ http://orcid.org/0000-0003-4121-7719
Jakub W Wojciechowski ⓘ http://orcid.org/0000-0001-5289-653X
Vince Grolmusz ⓘ http://orcid.org/0000-0001-9456-8876
Malgorzata Kotulska ⓘ http://orcid.org/0000-0002-2015-5339
Julie Deborah Forman-Kay ⓘ http://orcid.org/0000-0001-8265-972X
Peter J Roy ⓘ http://orcid.org/0000-0003-2959-2276

### Ethics

We (the authors) affirm that we have complied with all relevant ethical regulations for animal testing and research. Given that our experiments focused exclusively on the invertebrate nematode worm C. elegans, no ethical approval was required for any of the presented work.

### Decision letter and Author response

Decision letter https://doi.org/10.7554/eLife.79396.sa1
Author response https://doi.org/10.7554/eLife.79396.sa2

## Additional files

### Supplementary files

• Supplementary file 1. Genes of special interest and evidence of pharynx expression. Note that the data within this table is also available in the *Figure 4—source data 1* file so as to put it in

context with other data for each gene. Also note that the column indicators below are named after the column in Supplementary File 1. (C) All 78 gene products called out in *Figure 4A* are shown, along with all 226 genes represented in *Figure 4A*, in addition to 17 genes that are of special interest (including additional members of the *idpp* gene class referred to elsewhere in the text). (B) The APPGs that have higher sequence similarity to one another and have more similar temporal expression patterns are described as APPG family (#1) members to distinguish them from more divergent APPGs. (E) The Name Status indicates the 41 new WormBase-approved gene assignments. (H) The indicated hour and degree is with respect to *Figure 4A*. (J) In some cases, the updated Signal P algorithm will identify a signal peptide when ParaSite did not, as indicated with a 'no, but likley SS.' (L) The spatial expression patterns of the indicated clones can be inspected at http://nematode.lab.nig.ac.jp/dbest/srchbyclone.html. A green color indicates confirmation of the expected expression pattern (enriched in pharynx); 'no signal' indicates little to no signal anywhere in photo micrographs. In two cases indicated in pink, signal could be observed in the animal, but the pharynx lacked signal. (M, N) The PubMed ID number (PMID) is shown for the publication that provides additional spatial expression information for the gene. The nature of the data is either from a transgene (transg), an antibody (Ab), or is sequence-based (seq). A green color indicates confirmation of the expected expression pattern (enriched in pharynx).

• Supplementary file 2. Transcript levels of six low-complexity protein families within pharynx cells. The data is extracted from the data presented in *Figure 10B*.

• MDAR checklist

### Data availability
All source data for the spatiotemporal reconstruction is in the Source data files.

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
