## [Editor Report]

Cuticles are specialized extracellular matrices that cover the bodies of ecdysozoans, which make up 85% of all animals, and how cuticles are formed is very poorly understood, in particular in light of the fact that cuticles are shed and regrown as animals grow. The authors present a comprehensively and carefully curated resource of the components of the pharyngeal cuticle of *C. elegans* and provide a spatiotemporal framework to understand cuticle assembly. In doing so, the authors propose a function for a large class of intrinsically disordered proteins (IDPs). The significance of this work is high because our understanding of both cuticle formation and of IDPs is poor.

---

## [Decision Letter]

**Decision letter after peer review:**

Thank you for submitting your article "A Spatiotemporal Reconstruction of the *C. elegans* Pharyngeal Cuticle Reveals a Structure Rich in Phase-Separating Proteins" for consideration by *eLife*. Your article has been reviewed by 3 peer reviewers, one of whom is a member of our Board of Reviewing Editors, and the evaluation has been overseen by Piali Sengupta as the Senior Editor. The reviewers have opted to remain anonymous.

Essential revisions:

1) The reviewers appreciate that your analyses make strong predictions about the newly identified IDPs being incorporated into the pharyngeal cuticle. However, they also agree this needs to be supported by direct observation: i.e. generation and analysis of at least 2-3 fluorescent reporter lines (tagging at least one member of each of the newly described protein families). The endogenous tagging approach used for idpc-1 would be the preferred approach, as this will report endogenous expression patterns. But other approaches that similarly recapitulate endogenous expression may be acceptable.

---

## [Author Response]

Essential revisions:1) The reviewers appreciate that your analyses make strong predictions about the newly identified IDPs being incorporated into the pharyngeal cuticle. However, they also agree this needs to be supported by direct observation: i.e. generation and analysis of at least 2-3 fluorescent reporter lines (tagging at least one member of each of the newly described protein families). The endogenous tagging approach used for idpc-1 would be the preferred approach, as this will report endogenous expression patterns. But other approaches that similarly recapitulate endogenous expression may be acceptable.

Our new Figure 6 and respective text in the Results section addresses this issue. We now have made fusion proteins with a representative of all five relevant classes of proteins, including IDPA-3, IDPB-3, IDPC-1, NSPB-12, and FIPR-4. We also included additional analysis of ABU-14, a member of the APPG class of proteins. The expression of all six proteins is driven by their own promoter/enhancer elements upstream of their ATG. As described in David Raizen’s 2014 paper (PMID 25361578), the ABU-14 fusion protein is driven from a transgenic repetitive array. Save IDPC-1, in which we previously contracted InVivoBiosystems Inc to chromosomally tag, the expression of the remaining four reporters is driven with a transgenic repetitive array.

As Figure 6 shows, all six fusion proteins are expressed in association with the pharynx exclusively and localize to the pharynx cuticle. This conclusion is supported by the pattern of the tagged proteins localized to what is obvious cuticle structures (flaps, channel ribbing, grinder etc) and colocalization with the chitin stain Calcofluor White (CFW). What we find particularly striking is how closely the resulting patterns are predicted by the Cao and Packer et al. UMAP plots, which we also provide in Figure 6. Together with the other (multiple) lines of supporting evidence in the Results section titled ‘Orthogonal Data Validate the Spatiotemporal Map’**,** we hope these new observations satisfies the Reviewers and Editor.